# Structural basis of SNAPc-dependent snRNA transcription initiation by RNA polymerase II

Srinivasan Rengachari[1], Sandra Schilbach [1], Thangavelu Kaliyappan[2], Jerome Gouge[2], Kristina Zumer [1], Juliane Schwarz[3,4], Henning Urlaub [3,4], Christian Dienemann [1], Alessandro Vannini [2,5] ✉ & Patrick Cramer [1] ✉

RNA polymerase II (Pol II) carries out transcription of both protein-coding and non-coding genes. Whereas Pol II initiation at protein-coding genes has been studied in detail, Pol II initiation at non-coding genes, such as small nuclear RNA (snRNA) genes, is less well understood at the structural level. Here, we study Pol II initiation at snRNA gene promoters and show that the snRNA-activating protein complex (SNAPc) enables DNA opening and transcription initiation independent of TFIIE and TFIIH in vitro. We then resolve cryo-EM structures of the SNAPc-containing Pol II pre-initiation complex (PIC) assembled on U1 and U5 snRNA promoters. The core of SNAPc binds two turns of DNA and recognizes the snRNA promoter-specific proximal sequence element (PSE), located upstream of the TATA box-binding protein TBP. Two extensions of SNAPc, called wing-1 and wing-2, bind TFIIA and TFIIB, respectively, explaining how SNAPc directs Pol II to snRNA promoters. Comparison of structures of closed and open promoter complexes elucidates TFIIH-independent DNA opening. These results provide the structural basis of Pol II initiation at non-coding RNA gene promoters.

Transcription by RNA Pol II has been structurally well studied for protein-coding genes that produce messenger RNA (mRNA)[1-4]. Pol II, however, also carries out transcription of non-coding small nuclear RNAs (snRNAs) that are an integral part of the pre-mRNA splicing machinery[5]. Pol II transcribes four of the five snRNAs, namely U1, U2, U4 and U5 snRNAs, whereas Pol III transcribes U6 snRNA[6]. In contrast to the Pol III-dependent snRNA promoter, Pol II-dependent snRNA promoters lack a TATA box motif[7]. To produce snRNAs, Pol II uses many of its accessory factors that are used for mRNA synthesis, but it additionally requires specific factors for transcription initiation and elongation[8].

Transcription initiation of snRNA genes relies on a specific factor called snRNA-activating protein complex (SNAPc). SNAPc binds a DNA motif in the upstream region of snRNA promoters, referred to as the proximal sequence element (PSE)[9]. Human SNAPc contains five

subunits—SNAPC1, SNAPC2, SNAPC3, SNAPC4 and SNAPC5. The subunits SNAPC1, SNAPC3 and SNAPC4 form the core of SNAPc[10], of which SNAPC3 and SNAPC4 bind the promoter DNA[11,12]. The core subunits of SNAPc are conserved and have been characterized in *Drosophila melanogaster* and *Trypanosoma brucei*, in which they are sufficient for activating snRNA transcription[13,14]. SNAPC2 and SNAPC5, however, contribute to the stability and activity of SNAPc[10,15,16].

The initiation of SNAPc-regulated Pol II snRNA transcription has been reported to rely on the general transcription factors (GTFs) TBP, TFIIA, TFIIB, TFIIE and TFIIF[17,18]. The role of TFIIH in Pol II snRNA transcription remains unclear[17], although TFIIH is known to be required for DNA opening at promoters of protein-coding genes[19]. The structure of SNAPc and its molecular interactions with the Pol II pre-initiation complex (PIC) are also unknown. As a consequence, the structural

[1]Department of Molecular Biology, Max Planck Institute for Multidisciplinary Sciences, Göttingen, Germany. [2]Division of Structural Biology, The Institute of Cancer Research, London, UK. [3]Max Planck Institute for Multidisciplinary Sciences, Bioanalytical Mass Spectrometry, Göttingen, Germany. [4]University Medical Center Göttingen, Institute of Clinical Chemistry, Bioanalytics Group, Göttingen, Germany. [5]Human Technopole, Milan, Italy. ✉e-mail: alessandro.vannini@fht.org; patrick.cramer@mpinat.mpg.de

basis and the mechanism of snRNA transcription initiation remains to be uncovered. Here, we report structures of SNAPc-containing Pol II PICs bound to U1 and U5 snRNA promoters. Our results show how SNAPc is structured, how it recognizes the PSE and how it positions a core Pol II PIC on snRNA promoters for DNA opening and transcription initiation. More generally, this work adds to our understanding of the evolution of the three eukaryotic transcription systems.

## Results

### Preparation of functional SNAPc

We prepared two variants of recombinant human SNAPc: SNAPc-FL, which contains all full-length subunits, and SNAPc-core[10], which contains SNAPC1, SNAPC3, SNAPC4 (residues 1–516) and SNAPC5 (Fig. 1a) (Methods). Both purified SNAPc variants were able to bind U1 and U5 snRNA promoter DNA (RNU1 and RNU5), both in the absence and in the presence of TBP and TFIIB, in an electrophoretic mobility shift assay (EMSA) (Fig. 1b). The EMSA also showed that the SNAPc variants could facilitate binding of TBP to snRNA promoters that lack a TATA box (Figs. 1b and 2a), consistent with previous studies[20].

To test whether recombinant SNAPc could mediate Pol II transcription initiation from snRNA gene promoters, we used an in vitro transcription assay. The assay showed that Pol II could initiate transcription from a U1 promoter in the presence of TBP, TFIIA, TFIIB or TFIIF. The transcription was stimulated ~4.5 fold and ~2.5 fold in the presence of SNAPc-FL and SNAPc-core, respectively (Fig. 1d,e). Addition of TFIIE reduced this increase in transcription activity to ~1.5-fold and ~1.2-fold by SNAPc-FL and SNAPc-core, respectively (Fig. 1e). It is known that TFIIE has a stimulatory effect on transcription activity by Pol II[21,22]. However, the lack of functional cooperation between SNAPc and TFIIE suggests that TFIIE is not required for SNAPc-dependent transcription initiation and rather is inhibitory in our biochemical system. Further addition of TFIIH did override the stimulatory effect of SNAPc and led to an increase of non-specific transcripts at multiple sites (Fig. 1d). The increased background transcription could be a result of the DNA translocase activity of TFIIH that enables promoter opening at various DNA sites. In conclusion, our recombinant SNAPc variants stimulate Pol II transcription initiation from snRNA gene promoters in the absence of TFIIE and TFIIH in vitro.

### Cryo-EM analysis of SNAPc-containing PICs

On the basis of these observations, we reconstituted a functional SNAPc-containing Pol II PIC on a U1 promoter DNA (Methods). We incubated SNAPc-core and *Sus scrofa* Pol II (99.9% identical to human Pol II) with human TBP, TFIIA, TFIIB, TFIIE and TFIIF and subjected the resulting complex to sucrose-gradient ultracentrifugation. Peak fractions contained Pol II, SNAPc-core and GTFs in apparent stochiometry, indicating the formation of a stable SNAPc containing Pol II PIC (Fig. 1c). The sample was cross-linked[23] and subjected to cryo-EM analysis (Methods). Initial trials showed that the PIC containing the SNAPc-FL variant was less stable (Fig. 1c), whereas the PIC containing SNAPc-core was stable and suitable for cryo-EM analysis, leading to a high-resolution single-particle dataset (Table 1).

Reconstructions from 3D classification of this dataset showed two distinct particle classes of the SNAPc-containing Pol II PIC (Extended Data Fig. 1). Further 3D classification and refinement identified these two states as the closed promoter complex (CC) and the open promoter complex (OC) states of the PIC. We obtained structures of the CC and OC states at an overall resolution of 3.4 Å and 3.0 Å, respectively (Extended Data Figs. 1 and 3). None of our maps revealed density for TFIIE, consistent with our in vitro transcription assays showing that TFIIE was not required for initiation (Fig. 1d,e). The observed co-migration of SNAPc and TFIIE in the sucrose gradient assay hence results from the limited separation range of the experiment. Conversely, PIC species containing SNAPc or TFIIE were observed as two distinct particle classes at initial stages of cryo-EM data processing. Densities for SNAPc and upstream

DNA containing the PSE were improved by focused 3D classification and masked refinements. The local resolution for this region was 3.5 Å for the OC state (Extended Data Figs. 1 and 3).

To obtain a high-resolution structure of SNAPc, we additionally reconstituted a SNAPc-containing Pol II PIC on DNA that was based on the U5 promoter sequence (Methods). The resulting cryo-EM dataset enabled refinement of the SNAPc-containing PIC in the CC state at an overall resolution of 3.0 Å and with the local map of the upstream region extending to 3.2 Å (Extended Data Figs. 2 and 3). The local map enabled building of an atomic model for SNAPc and PSE-containing upstream DNA (Methods). We then combined the resulting model with the known high-resolution structures of mammalian Pol II PIC in the CC and OC states[24]. After manual adjustment, refined structures of the SNAPc-containing PIC in the CC and OC states containing the U1 and U5 promoters showed good stereochemistry, resulting in a total of three structures (Table 1).

### Overall structure of SNAPc-containing PIC

The overall structure of the SNAPc-containing Pol II PIC shows that SNAPc binds the promoter DNA upstream of TBP (Fig. 2). SNAPc recognizes the PSE motif and interacts with TFIIA and TFIIB. Despite these multiple interactions, the presence of SNAPc does not alter the canonical core PIC structure in any substantial way[24]. TBP binds to the AGGCTG sequence at register −30 to −25 bp (Fig. 2a) of the TATA-less U1 promoter and bends the DNA by 90°, similar to what is observed in a TBP–TATA DNA complex[25,26] (Extended Data Fig. 4a). In the following sections, we will first describe the SNAPc structure and SNAPc–DNA interactions on the basis of the U5-containing CC structure that is resolved at the highest resolution. We will then describe promoter opening on the basis of the CC and OC structures of the U1-containing PIC.

### SNAPc structure contains two protruding wings

The high-resolution structure of the SNAPc-core bound to the U5 promoter shows how the subunits SNAPC1, SNAPC3 and SNAPC4 fold and interact (Fig. 3). SNAPC1 possesses an amino-terminal VHS/ENTH-like domain[27] that forms a mainly helical structure (Extended Data Fig. 4b). SNAPC3 is saddle-shaped, with a central 'ubiquitin-like domain' (ULD) and additional α-helices and β-strands (Extended Data Fig. 4c). Consistent with biochemical studies[28], SNAPC3 contains two zinc fingers (ZF-1 and ZF-2). ZF-1 is a C2H2-type zinc finger, with residues Cys221, His313, Cys317 and His319 coordinating a $Zn^{2+}$ ion (Extended Data Fig. 5f). ZF-2 is a C4 type zinc finger, with residues Cys354, Cys357, Cys380 and Cys383 coordinating another $Zn^{2+}$ ion (Extended Data Fig. 5g). SNAPC4 contains four complete repeats (R1–R4) and a half repeat (Rh) of the Myb domain[12], of which we observed Rh, R1 and R2 (residues 274–398) (Fig. 3b and Extended Data Fig. 4d). R1 and R2 contain three helices forming canonical helix-turn-helix folds. The SNAPc-core is stabilized by intricate interactions of SNAPC3 with both SNAPC1 and SNAPC4. The N-terminal region of SNAPC3 interacts mainly with SNAPC1, burying a surface area of ~1,640 Å². The carboxy-terminal region of SNAPC3 binds SNAPC4 and buries ~3,010 Å². Four subunit interfaces are formed, based on hydrophobic interactions, salt bridges and polar contacts (Fig. 3c–f and Extended Data Figs. 7 and 8).

SNAPc also contains two protrusions that we refer to as 'wing-1' and 'wing-2.' Wing-1 of SNAPc consists of a pair of helices that precede the Rh region of SNAPC4 (residues 184–256). Wing-2 of SNAPc is a four-helix bundle that is formed by two helices of SNAPC1 (residues 162–234) and one helix each of SNAPC4 (residues 81–125) and SNAPC5 (residues 1–51) (Extended Data Fig. 4e). Although the resolution in wing-2 is limited owing to mobility, AlphaFold2 prediction[29] and prior biochemical studies[16] have led to a reliable model of wing-2 that we confirmed by cross-linking mass spectrometry (Extended Data Figs. 5k and 6). In conclusion, these efforts provided the structure of SNAPc, which contains a three-subunit core and two protruding wings extending from the core.

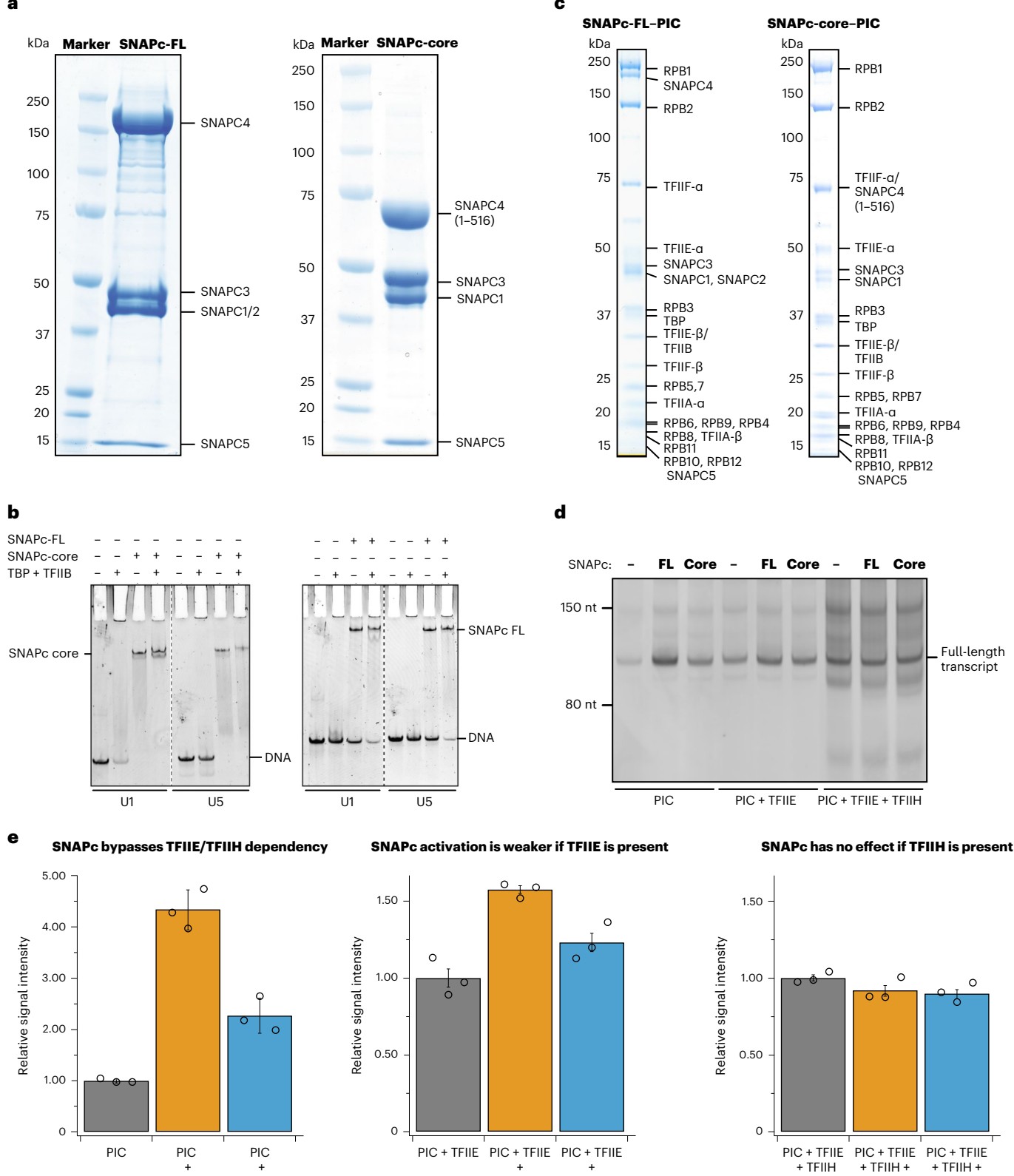

**Fig. 1 | Preparation of SNAPc-containing Pol II PIC on non-coding RNA promoters. a**, SDS–PAGE analysis of SNAPc variants (FL and core) purified to homogeneity. The experiments were repeated at least three times. **b**, EMSA showing the binding of SNAPc (with or without TBP and TFIIB) to U1 and U5 promoter DNA. The presence of SNAPc stabilizes the binding of TBP–TFIIB to snRNA promoters. The experiment was repeated at least three times. **c**, SDS–PAGE analysis of SNAPc-containing Pol II PIC variants isolated through sucrose gradient ultracentrifugation. The experiments were repeated at least three times.

**d**, In vitro transcription assay showing the relative influence of SNAPc variants on Pol II snRNA transcription with different combinations of GTFs. The gel is representative of triplicate experiments. **e**, Histogram plots representing the quantification (Methods) of full-length transcripts from the in vitro transcription assay in **d**. The presence of TFIIH with or without SNAPc leads to roughly four to seven fold increases in the formation of background RNA products. Data are presented as mean ± s.d. The error bars have been derived from three independent experiments.

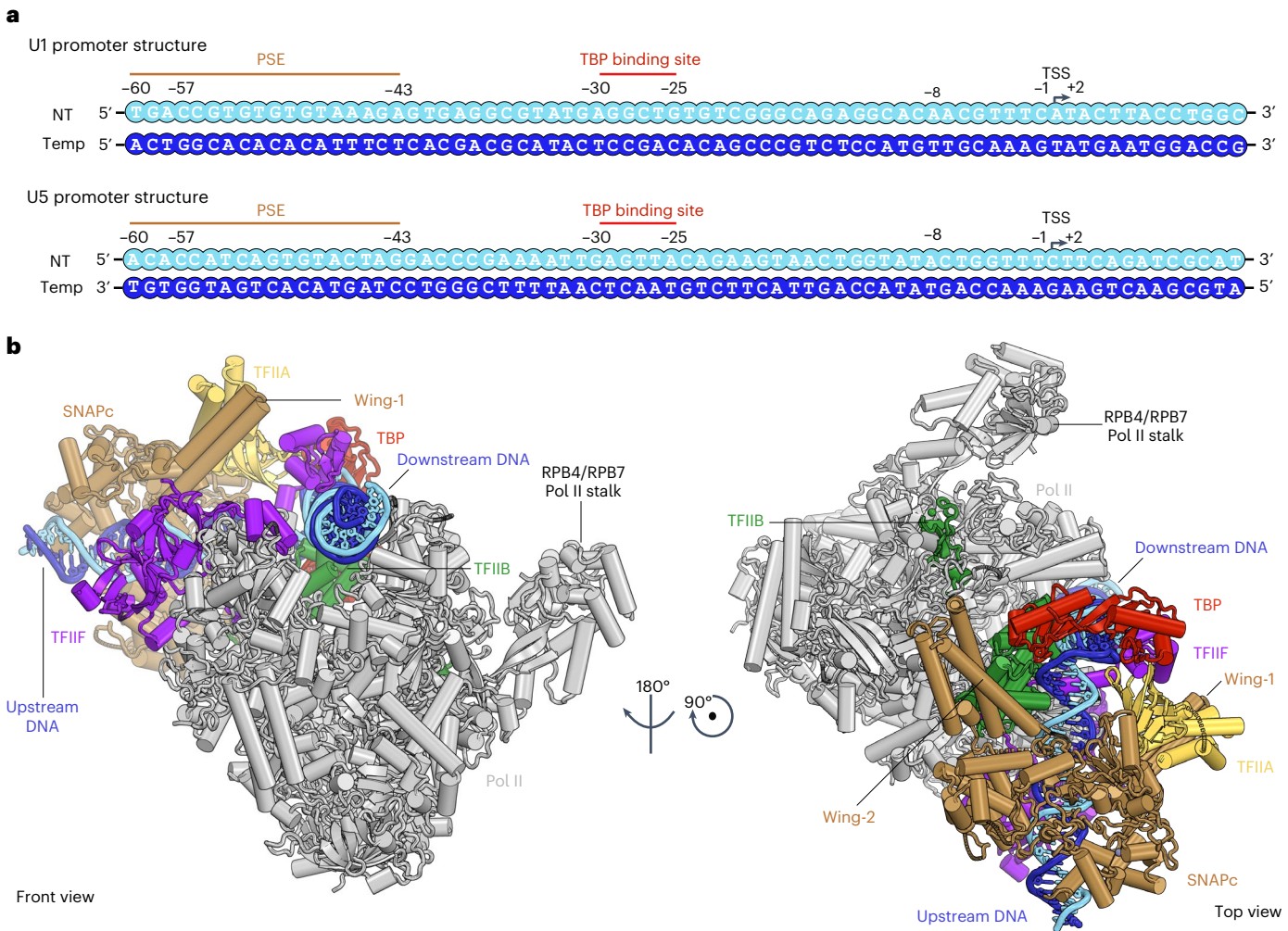

**Fig. 2 | Overall structure of SNAPc-containing Pol II PIC. a**, Schematic 2D representation of the U1 and U5 promoter sequences highlighting the binding motifs of the initiation machinery as observed in the cryo-EM structure: PSE (SNAPc), TBP binding site (TBP) and TSS (Pol II). The transcription start site (TSS) is denoted as +1, and negative and positive numbers indicate upstream and downstream positions. **b**, Cartoon representation of the SNAPc-containing Pol II PIC, as viewed from the front and top. The color codes for Pol II and the GTFs are consistently used throughout.

## SNAPc-core recognizes the snRNA promoter

Our U5-containing CC structure also reveals details of how SNAPc binds the PSE motif in promoter DNA (Fig. 4a). The SNAPc-core binds to the PSE motif through its subunits SNAPC3 and SNAPC4 (Extended Data Fig. 9a), consistent with biochemical data[11,30]. SNAPc contacts promoter DNA 8 bp upstream of the proximal edge of the TBP-binding site (Fig. 4b,c). The register of the modeled snRNA promoter is defined by the nucleotide on the non-template (NT) strand at the upstream edge of TBP binding site starting at −30 nt, ascending in the 5′ to 3′ direction. SNAPC3 and SNAPC4 bind both the major and minor grooves in this region through contacts with the DNA backbone and bases on both strands of the PSE (Extended Data Fig. 9a). SNAPC3 inserts its helix α8 into the major groove and forms multiple contacts with DNA. K199 forms salt bridges with the backbone phosphates of nucleotide G9 on the template strand. The residue K194 of the same helix forms ionic interactions with the O6 atom of the nucleotides G at positions −42 and −43 of the NT strand. Further downstream, H198 establishes hydrophobic contacts with the nucleotide T (position −45) on the non-template strand (Fig. 4b).

Because most of these protein–DNA contacts are to the DNA backbone, the question of how SNAPc recognizes the PSE arises. Our structure suggests that recognition is at least partially achieved by indirect readout. In particular, the DNA major groove is locally distorted at the PSE and differs from canonical B-DNA at registers −51 to −41 nt (Extended Data Fig. 9b). At the position where SNAP3 helix α8 is inserted into the major groove, the duplex geometry resembles A-form DNA[31] (Extended Data Fig. 9c). This deviation is also reflected in the minor grooves upstream and downstream of this site (Extended Data Fig. 9a,d).

SNAPc also binds the minor groove of DNA with subunits SNAPC3 and SNAPC4. Q152 of SNAPC3 a forms hydrogen bond with the nucleotide T (position −48) of the NT strand, and SNAPC4 residue Y372 interacts hydrophobically with the C3 atom of backbone sugar of the nucleotide A (position −50) of the template strand. Arginine residues R148 and R151 of SNAPC3 and R373 of SNAPC4 form salt bridges with the DNA backbone (Fig. 4c). Our structure also shows that the SNAPC4 Myb repeat R2 binds DNA via its helix α15 that contacts the anterior major groove, and early biochemical studies have indicated that the Myb repeats R3 and R4 are involved in DNA binding[12]. I388 establishes hydrophobic interactions with the nucleotide A (position −50) and the C5 atom of nucleotide C (position −51) on the template strand. The neighboring Y389 residue forms hydrogen bonds with the N7 atom of nucleotide A (position −55) and a hydrophobic interaction with nucleotide T (position −54) of the NT strand (Fig. 4c). The residues K347, R373 and R390 of SNAPC4 interact with the DNA backbone. Although biochemical studies had identified SNAPC3 and SNAPC4 as poor DNA

**Table 1 | Cryo-EM data collection, refinement and validation statistics**

| | U1-OC (EMDB-15009, PDB 7ZXE) | U1-OC, local map (EMDB-15007, PDB-7ZX8) | U1-CC (EMDB-15006, PDB-7ZX7) | U5-CC (EMDB-14997, (PDB-7ZWD) | U5-CC, local map (EMDB-14996, PDB- 7ZWC) |
|---|---|---|---|---|---|
| **Data collection and processing** | | | | | |
| Magnification | | ×81,000 | | ×81,000 | |
| Voltage (kV) | | 300 | | 300 | |
| Electron exposure (e⁻/Å²) | | 54.45 | | 51.93 | |
| Defocus range (μm) | | −0.5 to −3.0 | | −0.5 to −2.5 | |
| Pixel size (Å) | | 1.05 | | 1.05 | |
| Micrographs collected | | 16,854 | | 4,842 | |
| Initial particle images (no.) | | 5,181,947 | | 1,299,523 | |
| Final particle images (no.) | 137,246 | 137,246 | 47,293 | 85,787 | 85,787 |
| Map resolution (Å) | 3.0 | 3.5 | 3.4 | 3.0 | 3.2 |
| FSC threshold | 0.143 | 0.143 | 0.143 | 0.143 | 0.143 |
| Map resolution range (Å) | 2.8–5.2 | 3.4–6.0 | 3.0–7.0 | 2.75–5.25 | 3.0–5.0 |
| **Refinement** | | | | | |
| Initial model used (PDB code) | 7NVU | 7NVU | 7NVS | 7NVS | 7NVS |
| Map sharpening *B* factor (Å²) | −10 | −10 | −5 | −10 | −10 |
| Model composition | | | | | |
| DNA | 126 | 93 | 132 | 132 | 83 |
| Protein residues | 5,842 | 1,500 | 5,789 | 5,789 | 1,500 |
| Ligands | 11 | 2 | 12 | 12 | 2 |
| *B* factors (Å²) | | | | | |
| DNA | 276.52 | 174.43 | 318.34 | 248.19 | 95.70 |
| Protein residues | 133.43 | 186.65 | 215.69 | 124.47 | 109.50 |
| Ligands | 192.93 | 155.55 | 237.49 | 194.42 | 82.54 |
| R.m.s. deviations | | | | | |
| Bond lengths (Å) | 0.005 | 0.003 | 0.004 | 0.007 | 0.003 |
| Bond angles (°) | 0.755 | 0.606 | 0.509 | 0.658 | 0.524 |
| Validation | | | | | |
| MolProbity score | 1.78 | 1.68 | 1.69 | 1.63 | 1.67 |
| Clashscore | 9.52 | 8.46 | 9.85 | 7.55 | 7.63 |
| Poor rotamers (%) | 0.16 | 0.00 | 0.00 | 0.00 | 0.00 |
| CaBLAM outliers | 1.93 | 1.73 | 1.63 | 1.91 | 2.07 |
| Cβ outliers | 0.00 | 0.00 | 0.00 | 0.00 | 0.00 |
| Ramachandran plot | | | | | |
| Favored (%) | 95.91 | 96.61 | 97.03 | 96.68 | 96.27 |
| Allowed (%) | 4.09 | 3.39 | 2.97 | 3.32 | 3.73 |
| Disallowed (%) | 0.00 | 0.00 | 0.00 | 0.00 | 0.00 |

binders when investigated in isolation[10,11], our results suggest that formation of the SNAPc complex with its intricate interactions between these two subunits enables tight binding of the PSE, which explains how SNAPc recognizes the snRNA promoter.

**The wings of SNAPc bind TFIIA and TFIIB**
SNAPc also interacts with TFIIA and TFIIB that flank TBP in the PIC (Fig. 5 and Extended Data Fig. 9a). Whereas wing-1 of SNAPc binds to TFIIA, wing-2 binds TFIIB (Extended Data Fig. 9a). SNAPc interaction with TFIIA and TFIIB involves three interfaces that we call A, B and C. In interface A, the wing-1 of SNAPC4 (helices α4 and α5) slides under the four-helix bundle of TFIIA like a wedge, stabilizing the flexible bundle region (Fig. 5a). SNAPC4 additionally interacts with the β-barrel of TFIIA to form interface B (Fig. 5a). Interfaces A and B are formed by a combination of hydrophobic interactions, salt bridges and polar contacts. Incidentally, the TFIIA bundle has also been shown to interact with TAF4 and TAF12 in lobe B of the multisubunit TFIID complex that, like SNAPc, is important for promoter recognition[32]. Interface C is formed between wing-2 and the C-terminal cyclin fold of the TFIIB core (Fig. 5b). The wing-2 helices from SNAPC1 and SNAPC5 form contacts with the terminal α-helix of

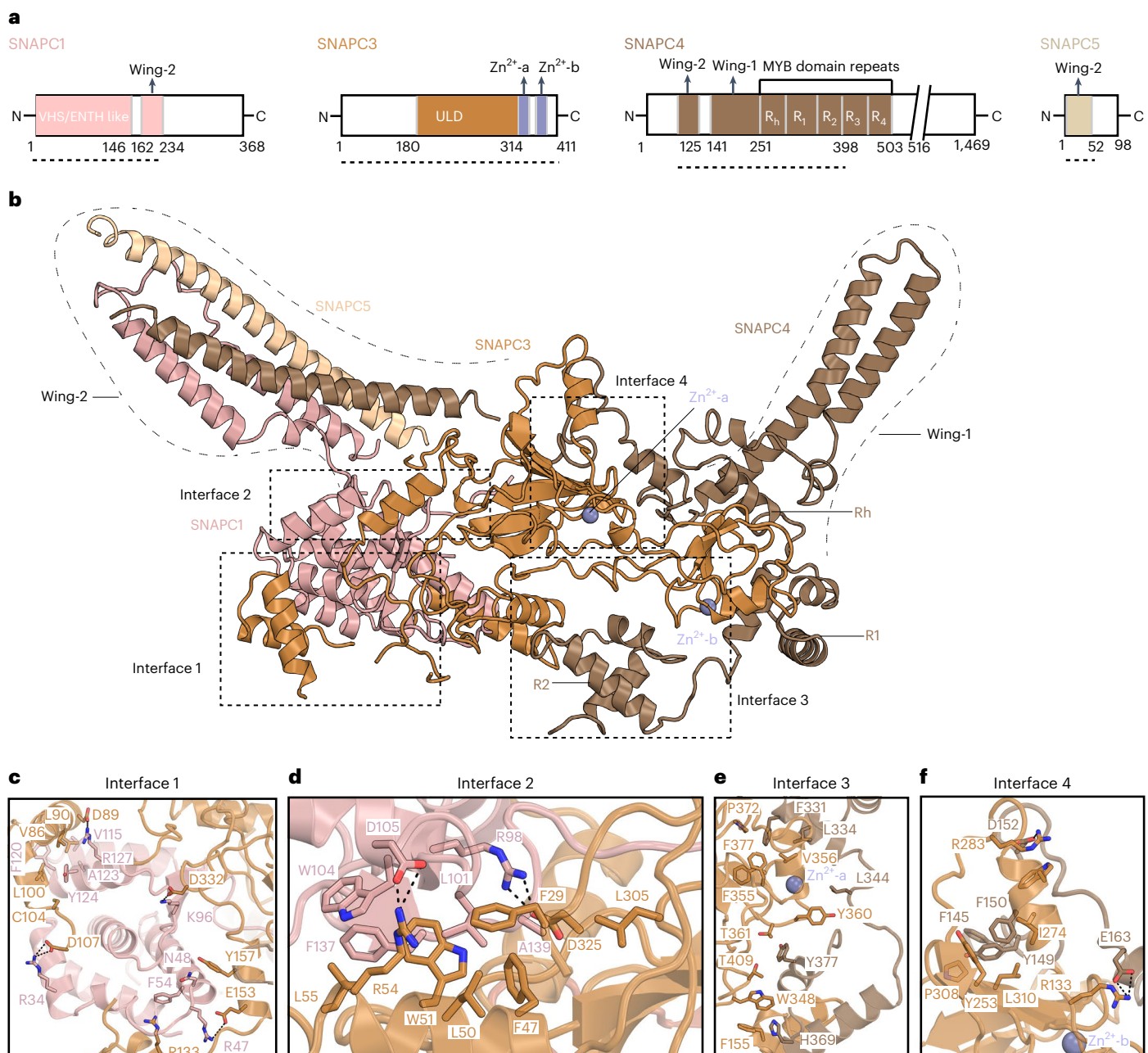

**Fig. 3 | Structure of SNAPc. a**, Two-dimensional (2D) domain schematics of individual SNAPc subunits. The regions visible in the 3D structure are marked by dotted lines. **b**, SNAPc structure in cartoon representation. Domain nomenclature and colors are as in **a**. Dashed boxes indicate the interfaces between the subunits. **c,d**, Close-up view of interfaces 1 and 2 that are formed between SNAPC1 (pink) and SNAPC3 (orange). The residues V115, F120, A123 and Y124 of SNAPC1 and V86, L90, L100 and C104 of SNAPC3 form mainly hydrophobic interactions, whereas ionic interactions are formed between R34, R47, K96 and R128 of SNAPC1 and D89, D107, E153 and D332 of SNAPC3. F54 of SNAPC1 and R133 of SNAPC3 form a cation-pi interaction, and N49 of SNAPC1 and Y157 of SNAPC3 form polar contacts. Similarly, in interface 2, SNAPC1 L101, W104, F137 and A139 form hydrophobic contacts with F47, L50, W51, L55 and L305

of SNAPC3. Salt bridges involving R98 and D105 of SNAPC1 and R54 and D325 of SNAPC3 fortify interface 2. **e,f**, Interfaces 3 and 4 between SNAPC3 (orange) and SNAPC4 (chestnut brown). In interface 3, SNAPC3 residues F155, W348, F355, V356, Y360, T361, P372, F377 and T409 form the bulk of hydrophobic contacts with F331, L334, L344 and H369 of SNAPC4 (Fig. 3e). Likewise, in interface 4, the residues Y253, I274, W277, P308 and L310 make hydrophobic contacts with the amino acids F140, Y149, F150 and F176 of SNAPC4. Additional salt bridges are formed by R133 and R283 of SNAPC3 with D152 and E153 of SNAPC4. The Zn-fingers (ZF-1 and ZF-2) of SNAPC3 are in close proximity to the interfaces 3 and 4 and would be important for the structural integrity of this complex. The residues involved in these protein–protein interaction surfaces are highly conserved across metazoans (Extended Data Figs. 7 and 8).

the TFIIB core. Interface C stabilizes the TFIIB core, which has been suggested to have a key role in the activation of snRNA transcription initiation[7]. Together, SNAPc wing-1 and wing-2 bind TFIIA and TFIIB, respectively, to position the core PIC with respect to SNAPc and the PSE promoter element.

## Promoter DNA opening

Comparison of our CC and OC structures bound to the U1 promoter provides insights into the mechanism of TFIIE- and TFIIH-independent DNA opening (Fig. 6). Overall, closed and open U1 promoter DNA follows trajectories within the Pol II cleft that are comparable to those observed for

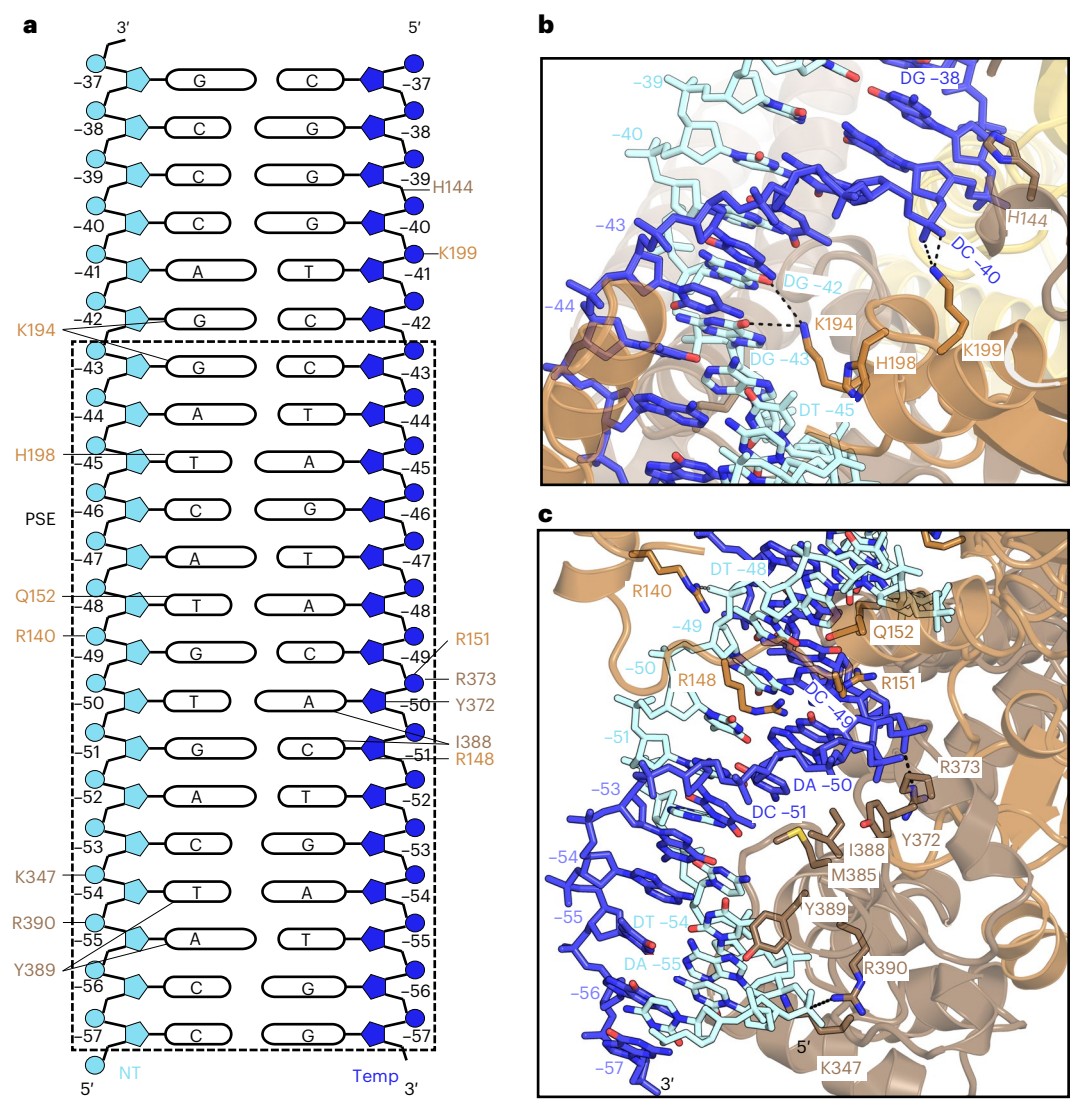

**Fig. 4 | SNAPc–DNA interactions. a**, Schematic view of the protein–DNA interactions between SNAPc and the PSE motif. Residues interacting with specific regions of the DNA, as described in the text, are indicated by lines. In panels **b** and **c**, the DNA is labelled and colour coded to indicate the register and the strands, respectively. **b**, DNA–protein interaction network on the preceding major and minor grooves (register: −46 to −35) of PSE, as bound by SNAPc subunits SNAPC3 and SNAPC4. Color codes are used uniformly in all panels. **c**, Close-up view of the first major and minor groove (register: −57 to −47) interactions between SNAPc and the PSE motif on the U5 promoter. The SNAPc subunits are represented as a cartoon, whereas the interacting amino acid side chain residues and DNA chains are depicted as sticks with atomic colors. Dashed lines indicate ionic interactions.

protein-coding promoter DNA in the PIC structure[24]. Also, as observed in PIC structures lacking SNAPc[2,24], the OC state is associated with a closed Pol II clamp and an ordered B-reader and B-linker elements in TFIIB (Fig. 6b). However, DNA opening can also be achieved spontaneously in the absence of TFIIE and TFIIH at some protein-coding genes in yeast[22], and such spontaneous opening depends on the DNA duplex stability around the transcription start site (TSS)[33]. Studies in yeast Pol II have further shown that an AT-rich sequence increases the propensity of spontaneous promoter opening during transcription intiation[22]. Similarly, we find that promoter sequences of snRNA-encoding genes are AT-rich in the initially melted region (IMR) that spans positions −8 to +2 around the TSS (position +1) (Extended Data Fig. 9e). We propose that the AT-rich nature of the IMR enables spontaneous DNA opening of the U1 promoter upon PIC binding. In summary, these results suggest that DNA opening of snRNA gene promoters and the spontaneously melted protein-coding genes rely on easily melting regions around the TSS and use similar mechanisms.

## Definition of the transcription start site

We observe 19 nucleotides (nt) of the DNA template strand spanning from the TBP-binding site to the upstream edge of the DNA bubble (at position −12). The templating nucleotide in open promoter DNA reaches the active site of Pol II ~30 nt downstream of the upstream edge of the TBP-binding site (Fig. 6a,b). The DNA strands forming the open DNA bubble are mobile, leading to a weakly resolved map. Subsequently, 12 nt further downstream, we observe T (position +1) of the template strand immediately downstream of the catalytic $Mg^{2+}$ ion at the active site. This posits residue G (position −1) as the template for RNA synthesis. The CA dinucleotide is the signature of the Initiator sequence (Inr)[34] and is located at register −1 and +1 of the non-template strand. This observation suggests that the TSS position is defined by a fixed distance from the site of TBP binding, as is known for protein-coding human genes that have their TSS within a window of 28–33 bp downstream of the TATA box[35]. Because we also observe a fixed position of

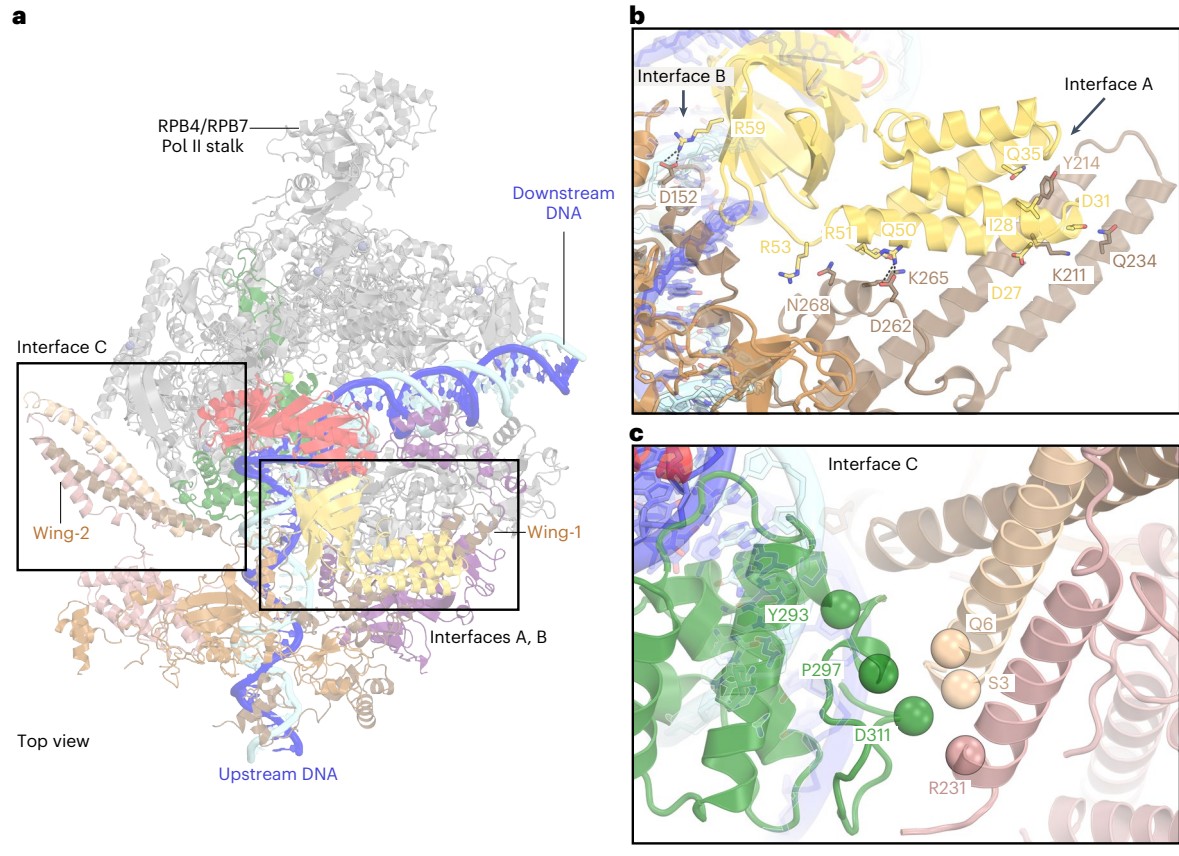

Pol II TBP TFIIA TFIIB SNAPC1 SNAPC3 SNAPC4 SNAPC5 NT Temp

**Fig. 5 | SNAPc–general transcription factors interaction. a**, Structure of SNAPc-containing Pol II PIC, with subunits of SNAPc colored as in Figure 3. **b**, Close-up view of the wing-1–TFIIA interaction. The amino acid residues involved in the formation of interfaces A and B between TFIIA and SNAPC4 are represented as sticks. Dashed lines indicate salt-bridges. **c**, Zoomed-in view of the interface C formed between wing-2 and TFIIB C-terminal cyclin fold. The Cα atoms of putative residues forming the interaction surface are represented as spheres.

SNAPc with respect to TBP, the TSS is apparently set by a fixed distance from the PSE in snRNA promoters.

These observations suggest that Pol II transcription would initiate from a TSS that is rather precise in vivo. To investigate this, we identified the main TSSs and determined their 'TSS precision scores' from a reanalysis of 5′-capped RNA-sequencing data[36] for both mRNA- and snRNA-encoding genes with a constitutive first or a single exon (Methods). A maximum precision score of 1 means that all transcripts initiate at the main TSS (±2 bp). Indeed, we find that Pol II snRNA transcription generally initiates in this narrow, 5-bp window, with a high median precision score of 0.86, as exemplified by the *RNVU1-15* promoter (Fig. 6c). In contrast, Pol II initiates transcription less precisely at TATA-less mRNA promoters, as shown by a median precision score of 0.36 and exemplified by the *HAT1* promoter. Pol II also initiates mRNA transcription more precisely when promoter DNA contains a TATA box motif, with a median precision score of 0.71, as exemplified by the *TUBB4B* promoter (Fig. 6c). These large differences in TSS precision are also observed in genome browser views of representative promoters (Fig. 6d). The observed high TSS precision of snRNA promoters is consistent with our model, in which SNAPc defines TSS position. In summary, SNAPc binding to the PSE likely serves as a ruler for positioning of TBP at TATA-less snRNA promoters, leading to initiation at a defined distance downstream of the PSE.

## Discussion

Here we report structures of SNAPc-containing Pol II PICs on two different snRNA gene promoters and in two different states, the CC and OC states. Together with biochemical results and published literature, our structures suggest the mechanism of SNAPc-mediated snRNA transcription initiation by Pol II (Fig. 7). SNAPc uses its conserved core to recognize the PSE motif in snRNA promoters, whereas its two wings position TFIIA and TFIIB. Since TFIIA and TFIIB form a rigid complex with TBP, SNAPc can indirectly position TBP at a defined location on snRNA promoters despite the absence of a consensus TATA box motif. This is consistent with the evidence that TFIIB–TBP complexes can be effectively recruited to snRNA promoters exclusively as part of a ternary TFIIA–TFIIB–TBP complex[18]. Positioning of the TFIIA–TFIIB–TBP complex on promoter DNA in turn recruits the Pol II–TFIIF complex to the IMR of the promoter. The low DNA duplex stability at the IMR enables spontaneous DNA opening and occurs with the use of binding energy independent of TFIIE and TFIIH. The emerging DNA template strand then binds in the Pol II active center cleft, and RNA chain synthesis is initiated at an Inr dinucleotide CA[34], thereby setting the TSS at a defined distance from the PSE.

Comparison of our results with published data also provides insights into the evolution of the three eukaryotic transcription systems. A distinguishing feature of transcription initiation by Pol II, with respect to Pol I and Pol III, is that the latter two machineries can open promoter DNA spontaneously[37–41], whereas Pol II machinery generally requires the help of an ATP-dependent translocase subunit in TFIIH and its accessory factor TFIIE[24,42]. However, we show here that, on snRNA promoters, mammalian Pol II, together with the factors that form the core PIC, can open DNA spontaneously without the help of TFIIE and TFIIH. Such spontaneous DNA opening

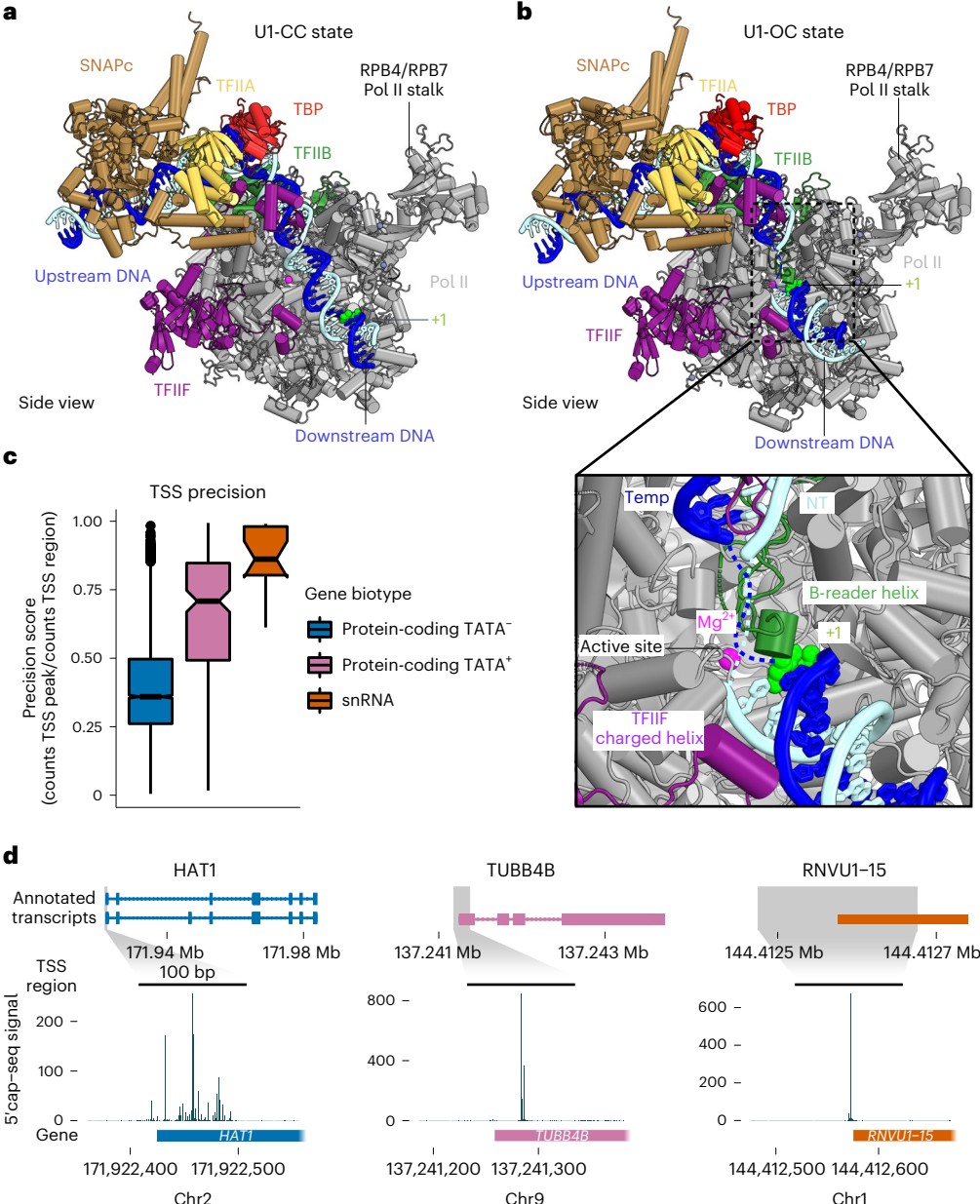

**Fig. 6 | Promoter opening. a**, Structure of SNAPc-containing Pol II PIC bound to U1 promoter in closed promoter complex (CC) state. The subunits are colored as in Figure 2. The nucleotide residue at the TSS (+1) on the template strand (blue) is represented as spheres (green). The Pol II active site metal ion A is depicted as a magenta sphere. **b**, Structure of SNAPc-containing Pol II PIC bound to U1 promoter in open promoter complex (OC) state. The inset represents a zoom into the active center containing open promoter DNA. The catalytic $Mg^{2+}$ ion at the active site is represented as a magenta sphere. The B-reader helix of TFIIB and the charged helix of TFIIF are highlighted alongside the +1 nucleotide residue represented as a sphere (green). **c**, Box plots showing TSS precision of protein-coding and snRNA genes ($n = 18$) transcribed by Pol II in cells. Protein-

coding genes are sub-grouped on the basis of promoter sequence into TATA-less (TATA⁻, $n = 4,521$) and TATA-containing (TATA⁺, $n = 200$) subsets. The thickened line represents the median value, the hinges correspond to the first and third quartiles, and the notches extend to 1.58 times the inter-quartile range divided by the square root of $n$. The whiskers represent the largest or smallest value within 1.5 times the inter-quartile range from the hinge, and outliers are shown in black. The precision scores were determined from published 5′ cap-seq data[36] (Methods). **d**, Annotated transcripts of representative examples from subsets in **c**, and genome browser views showing the 5′ cap-seq signal in the magnified region (±100 bp), centered at the main TSS peak. The annotated gene region is shown below the views, and only the sense strand signal is shown.

has also been observed for yeast Pol II at a subset of promoters[22] and also in the related archaeal transcription system[43]. Whereas spontaneous DNA opening occurs in the upstream-to-downstream direction, TFIIH-assisted DNA opening occurs in the downstream-to-upstream direction[24,42]. Our work thus provides evidence that, depending on the promoter, Pol II can use both types of DNA-opening mechanisms and suggests that TFIIH-assisted DNA

opening originated later in the evolution of cellular DNA-dependent RNA polymerase machinery.

Several open questions remain to be addressed for a better understanding of snRNA gene transcription. In particular, it has been found that SNAPc is regulated by its direct interaction with activators that localize ~200-bp upstream of the PSE at the distal sequence element (DSE)[7]. The intervening genomic region between the PSE and

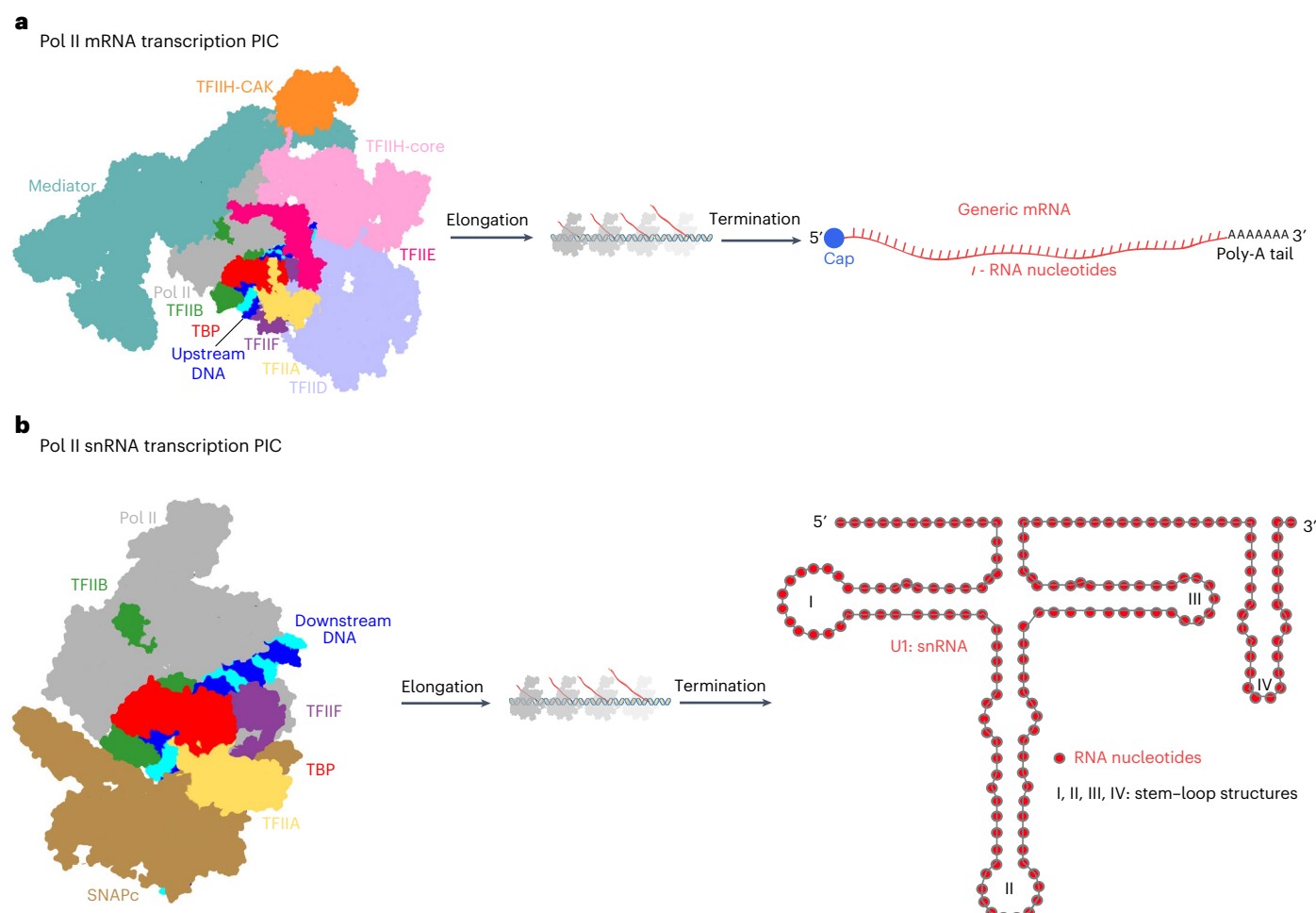

**Fig. 7 | Comparisons of Pol II PICs for mRNA and snRNA synthesis. a**, The Pol II PIC on protein-coding genes, bound to its elaborate array of initiation factors such as TFIIA, TFIIB, TFIID, TFIIE, TFIIF, TFIIH and Mediator complex. **b**, The Pol II PIC for snRNA transcription requires SNAPc, but not TFIIE, TFIIH or Mediator, to initiate transcription.

DSE may be decorated by a nucleosome[8]. In the future, our work may be expanded to study how DSE binding activators interact with the SNAPc-containing Pol II PIC described here and how a nucleosome may enable or modulate this interaction. Additionally, our work also serves as a stepping stone towards addressing the function of SNAPc in U6 snRNA transcription by Pol III. Such work should provide insights into how SNAPc can interact with both the Pol II and the Pol III initiation machinery, providing further insights into the evolution of eukaryotic transcription systems.

## Online content

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

## Methods

### Cloning and protein expression

cDNA constructs of SNAPc-FL containing SNAPC4 with an N-terminal StrepTwin-tag and a C-terminal His-tag, SNAPC1, SNAPC2, SNAPC3 and SNAPC5 were subcloned into the pLIB vector. The genes were assembled into a pBIG2ab vector using the biGBac system[44]. The cloned construct was transformed into DH10 EMBacY cells to generate bacmids. Next, the purified bacmid was mixed with Cellfectin II reagent (Thermo Fisher Scientific) and transfected into 2 ml (density: 0.5 million cells/ml) of adherent Sf9 cells in a 6-well plate. After incubating the plate at 27 °C for 72 hours, the resulting supernatant (P1 virus) was collected. To amplify the viral stock, 2 ml of P1 virus was added to 25 ml of Sf9 cells (0.5 million cells/ml) and incubated at 27 °C with shaking at 130 r.p.m. The supernatant (P2 virus) was collected after 4–5 days of infection, when the cell viability dropped to <85%, and was stored at 4 °C. Large-scale protein expression was carried out using 3 × 400 ml of High5 cells (0.5 million cells/ml) by adding 2 ml of P2 virus in each flask and incubating the flasks at 27 °C for 4 days at 130 r.p.m. Cells were then collected by centrifugation at 250$g$ for 10 minutes at 4 °C, and pellets were stored at −80 °C. SNAPc-core (SNAPC4 1-516 and lack of the SNAPC2 subunit) was expressed as previously described[18].

### Protein purification

The insect cells pellet of SNAPc-FL were resuspended in buffer A containing 50 mM HEPES pH 7.8, 750 mM NaCl, 10% glycerol, 15 mM imidazole, 10 mM β-mercaptoethanol, 2 mM MgCl$_2$, 1 mM phenyl-methylsulfonyl fluoride (PMSF), 1 μg/mL aprotinin, 1 μg/mL pepstatin and 1 μg/mL leupeptin, supplemented with four EDTA-free protease inhibitor tablets (Pierce), DNAse I (50 μg/ml) and 10 μl benzonase. Lysis was performed using a dounce homogenizer followed by sonication, and the lysate was clarified by centrifugation at 48,000$g$ at 4 °C for 40 minutes. The supernatant was filtered using a 0.45-μm filter and applied onto a HisTrap HP 5 ml column (GE Healthcare), pre-equilibrated with buffer A. The column was washed with 10 CV of buffer A1 (50 mM HEPES pH 7.8, 500 mM NaCl, 10% glycerol, 50 mM imidazole, 10 mM β-mercaptoethanol, 0.5 mM PMSF and 10 mM $O$-phospho-L-serine) and then with 5 CV of buffer A2 (50 mM HEPES pH 7.8, 1,250 mM NaCl, 10% glycerol, 50 mM imidazole, 10 mM β-mercaptoethanol and 0.5 mM PMSF). The column was again washed with 5 CV buffer A1, and the bound protein complex was eluted in buffer B (50 mM HEPES pH 7.8, 500 mM NaCl, 10% glycerol, 300 mM imidazole, 10 mM β-mercaptoethanol and 0.5 mM PMSF). Next, the sample was diluted to 250 mM NaCl with buffer heparin A (50 mM HEPES pH 7.8, 10% glycerol, 1 mM TCEP and 0.1 mM PMSF). The sample was centrifuged at 13,000 r.p.m. for 15 minutes at 4 °C and loaded onto a HiTrap Heparin HP 5-ml column (GE healthcare), pre-equilibrated with 12.5% of buffer heparin B (50 mM HEPES pH 8, 2 M NaCl, 10% glycerol, 1 mM TCEP and 0.1 mM PMSF). After washing with 5 CV of 12.5% buffer heparin B, elution was performed through a linear gradient from 15% to 60% over 10 CV. The eluted fractions were analyzed by SDS–PAGE, and fractions containing the SNAPc-FL complex were pooled, and concentrated using a 100-kDa molecular weight cut-off (MWCO) VivaSpin concentrator (Sartorius). The sample was centrifuged at 13,000 r.p.m. for 15 minutes at 4 °C and applied onto a Superose 6 PG XK 16/70 column (GE Healthcare), pre-equilibrated with 50 mM HEPES pH 7.8, 250 mM KCl, 10% glycerol and 1 mM TCEP. Peak fractions were pooled, concentrated, flash-frozen and stored at −80 °C.

SNAPc-core was purified as previously described[18], with some modifications. Briefly, after cell lysis and centrifugation, the supernatant was subjected to nickel column purification (GE Healthcare) and eluted with 300 mM imidazole. The elution was then further purified with an heparin column and eluted with a gradient from 250 mM to 1.25 M NaCl. The fractions of interest were pooled, concentrated and subjected to size-exclusion chromatography with a S200 16/600 equilibrated with 100 mM NaCl, 50 mM HEPES pH 7.9, 10% glycerol

and 1 mM TCEP. *S. scrofa* Pol II and human initiation factors TBP, TFIIA, TFIIB, TFIIE, TFIIF and TFIIH were purified as previously described[24].

### Electrophoretic mobility shift assays

EMSA was performed using a 76-bp fragment of U1 promoter DNA (template: 5′-GAA ACG TTG TGC CTC TGC CCC GAC ACA GCC TCA TAC GCC TCA CTC TTT ACA CAC ACG GTC ACT TG CCC CGC GCA CT-3′ and its complementary strand) and a 75-bp fragment of U5 promoter DNA (template: 5′-ACC AGT TAC TTC TGT AAC TCA ATT TTC GGG TAA CTG CAA TTC CTA GTA CAC TGA TGG TGT CTA CTA ATC CC AAG G-3′ and its complementary strand; Integrated DNA Technologies). First, 20 pM of SNAPc-FL or core was incubated with 5 pM of annealed oligonucleotides in the presence or absence of 25 pM of TFIIB and TBP in 20 μl of incubation buffer (250 mM NaCl, 50 mM HEPES pH 7.9, 20% glycerol, 1 mM TCEP) at room temperature for 15 minutes. The complexes were resolved on 5% polyacrylamide (37.5:1 acrylamide/bis-acrylamide, 10% glycerol, Tris Borate EDTA 1×) gels in 0.5× Tris Borate EDTA running buffer at 40 mA. After staining with ethidium bromide, the gels were scanned with a Typhoon FLA9500 (GE Healthcare).

### Promoter-dependent in vitro transcription assay

In vitro transcription assays were performed as described previously[24,42] with minor alterations. The DNA scaffold (dsDNA) was prepared as reported using a pUC119 vector into which a 92-nt fragment of the native U1 snRNA promoter[20] had been inserted. The scaffold (non-template: 5′-GGG CGT GAC CGT GTG TGT AAA GAG TGA GGC GTA TGA GGC TGT GTC GGG GCA GAG GCA CAA CGT TTC GCC CGA AGA TCT CAT ACT TAC CTG GCA GGG CTA AGC TTG GCG TAA TCA TGG TCA TAG CTG TTT CCT GTG TGA AAT TGT TAT CCG CTC ACA ATT CCG CCC-3′, template: 5′-GGG CGG AAT TGT GAG CGG ATA ACA ATT TCA CAC AGG AAA CAG CTA TGA CCA TGA TTA CGC CAA GCT TAG CCC TGC CAG GTA AGT ATG AGA TCT TCG GGC GAA ACG TTG TGC CTC TGC CCC GAC ACA GCC TCA TAC GCC TCA CTC TTT ACA CAC ACG GTC ACG CCC-3′) was stored in low-salt buffer (60 mM KCl, 10 mM K-HEPES pH 7.5, 8 mM MgCl$_2$, 3% (vol/vol) glycerol).

Initiation complexes for in vitro transcription were reconstituted on scaffold DNA essentially as has been described[24,42]. All incubation steps were performed at 25 °C, unless indicated otherwise. Per sample, 1.6 pmol scaffold, 1.8 pmol Pol II, TFIIE and TFIIH, 5 pmol TBP and TFIIB, 9 pmol TFIIF and TFIIA and 5 pmol SNAPc-FL or SNAPc-core were used. SNAPc was mixed and added to the sample simultaneously with TFIIB. Reactions were prepared in a sample volume of 23.8 μl, with final assay conditions of 60 mM KCl, 3 mM K-HEPES pH 7.9, 20 mM Tris-HCl pH 7.9, 8 mM MgCl$_2$, 2% (wt/vol) PVA, 3% (vol/vol) glycerol, 0.5 mM 1,4-dithiothreitol, 0.5 mg/ml BSA and 20 units RNase inhibitor. To achieve complete PIC formation, samples were incubated for 45 minutes at 30 °C. Transcription was started by adding 1.2 μl of 10 mM NTP solution and permitted to proceed for 60 minutes at 30 °C. Reactions were quenched with 100 μl Stop buffer (300 mM NaCl, 10 mM Tris-HCl pH 7.5, 0.5 mM EDTA) and 14 μl 10% SDS, followed by treatment with 4 μg proteinase K (New England Biolabs) for 30 minutes at 37 °C. RNA products were isolated from the samples as described[42], applied to urea gels (7 M urea, 1× TBE, 6% acrylamide:bis-acrylamide 19:1) and separated by denaturing gel electrophoresis (urea–PAGE) in 1× TBE buffer for 45 minutes at 180 volts. Gels were stained for 30 minutes with SYBR Gold (Thermo Fisher Scientific), and RNA was visualized with a Typhoon 9500 FLA imager (GE Healthcare Life Sciences).

### Preparation of the SNAPc-containing Pol II PIC

We performed the assembly of SNAPc-containing Pol II PIC on snRNA promoters at 25 °C, essentially as described previously. We used a 96-bp fragment of both the native U1 promoter DNA (template: 5′-ATC ATG GTA TCT CCC CTG CCA GGT AAG TAT GAA ACG TTG TGC CTC TGC CCC GAC ACA GCC TCA TAC GCC TCA CTC TTT ACA CAC ACGGTC ACT TGC-3′; non-template: 5′-GCA AGT GAC CGT GTG TGT AAA GAG TGA

GGC GTA TGA GGC TGT GTC GGG GCA GAG GCA CAA CGT TTC ATA CTT ACC TGG CAG GGG AGA TAC CAT GAT-3') and an engineered U5 promoter with 10 bp deleted from the downstream edge of the PSE sequence (template: 5'- CCC TGC CAG GTT TTA TGC GAT CTG AAG AGA AAC CAG AGT ATA CCA GTT ACT TCT GTA ACT CAA TTT TCG GGT CCTAGT ACA CTG ATG GTG TCT ACT-3'; non-template: 5'-AGT AGA CAC CAT CAG TGT ACT AGG ACC CGA AAA TTG AGT TAC AGA AGT AAC TGG TAT ACT CTG GTT TCT CTT CAG ATC GCA TAA AAC CTG CAG GGG-3'). In summary, SNAPc (FL or Core) was pre-incubated for 5 minutes with the snRNA promoter (U1 or U5) scaffold. It was then mixed with TFIIA−TFIIB and TBP, followed by the pre-formed Pol II−TFIIF complex. TFIIE was then added to this mixture, and the assembly was incubated at 25 °C for 60 minutes at 300 r.p.m. This reconstituted SNAPc-containing Pol II PIC was subjected to 10−30% sucrose-gradient ultracentrifugation with simultaneous cross-linking using GraFix (Kastner et al.[23]) at 175,000$g$ for 16 hours at 4 °C. The assay was then fractionated as 200-µl aliquots, where the cross-linking reaction was quenched using a cocktail of 10 mM aspartate and 30 mM lysine for 10 minutes. Fractions with SNAPc-containing Pol II PIC were dialyzed against the cryo-EM sample buffer (25 mM HEPES pH 7.6, 100 mM KCl, 5 mM MgCl$_2$, 1% glycerol and 3 mM TCEP).

## Cryo-EM data collection and processing
Samples for cryo-EM were prepared using Quantifoil R3.5/1 holey carbon grids pre-coated with a homemade 3 nm continuous carbon. Four microliters of SNAPc-containing Pol II PIC sample bound to snRNA promoter (U1/U5) were added to the carbon side and incubated for 2.5 minutes. The grids were blotted for 2.5 seconds and vitrified by plunging into liquid ethane with a Vitrobot Mark IV (FEI Company) set at 4 °C and 100% humidity. Cryo-EM data were collected on a 300-kV FEI Titan Krios with a K3 summit direct detector (Gatan) and a GIF quantum energy filter (Gatan) operated with a slit width of 20 eV. Automated data collection was performed with SerialEM at a nominal magnification of ×81,000, corresponding to a pixel size of 1.05 Å/pixel[45]. For the sample containing the U1 promoter, 16,854 image stacks, with each stack containing 50 frames, were collected at a defocus range of −0.5 to −3.0 µm. All movie frames were contrast transfer function (CTF)-estimated, motion-corrected and dose-weighted using Warp[46]. Particles were picked by Warp using a trained neural network, resulting in 5,181,947 particles as a starting set. Subsequent steps of image processing were performed with cryoSPARC[47] and RELION v.3.1.0 (ref. [48]).

Particles were extracted with a binning factor of 2 and a box size of 200 pixels (a pixel size of 2.1 Å/pixel) to perform initial clean-up and sorting. The processing scheme was centered around identifying the best SNAPc-containing particle sets. Iterative rounds of 2D classification, followed by heterogenous and homogenous refinements in cryoSPARC, led to two sets of particles corresponding to CC (set 1: 252,067 particles) and OC (set 2: 240,243 particles) promoter states, respectively. Each set was re-extracted without binning and processed using RELION v.3.1.0, as follows. For set 1, the particles were further sorted by focused 3D classification with a large spherical mask (mask 1) encompassing the upstream region of PIC containing SNAPc, TBP, TFIIA and TFIIB. This resulted in identifying the best 47,293 SNAPc-containing particles. These particles were again subjected to 3D refinement using mask 1, giving rise to a reconstruction of SNAPc-containing Pol II PIC bound to U1 promoter in CC state at 3.4-Å resolution (map 1). In parallel, focused 3D classification of set 2 with a spherical mask (mask 2) around the upstream region helped to identify the best 137,246 SNAPc-containing particles. These particles were then subjected to 3D refinement, followed by CTF refinement and Bayesian polishing. Following this, the particles were refined with and without mask 1 to obtain of SNAPc-containing Pol II PIC bound to U1 promoter in the OC state at 3.0 Å (map 2) and a local map spanning the SNAPc-containing upstream region at 3.7-Å resolution (map 3).

For the sample containing the U5 promoter, 4,842 image stacks, with each stack containing 60 frames, were collected at a defocus range of −0.3 to −2.5 µm. All movie frames were contrast transfer function (CTF)-estimated, motion-corrected and dose-weighted using Warp[46]. Particles were picked by Warp using a trained neural network, resulting in 1,299,523 particles. Subsequent image-processing steps were performed with cryoSPARC[47] and RELION v.3.1.0 (ref. [48]). Particles were extracted with a binning factor of 4 and a box size of 100 pixels (a pixel size of 4.2 Å/pixel) to perform initial clean-up and sorting. After sorting in cryoSPARC using 2D classification followed by heterogenous and homogenous refinements, a particle set (set 3: 443,960 particles) in CC promoter state was re-extracted with 2× binning (a pixel size of 2.1 Å/pixel) and processed using RELION v.3.1.0, as follows. For set 3, the particles were further sorted by 3D classification, followed by focused 3D classification using mask 1. The resulting 159,144 particles were re-extracted without a binning factor and were subjected to CTF refinement and Bayesian polishing. These particles were then subjected to another round of masked classification, yielding 85,787 SNAPc-containing particles. These particles were then 3D refined with and without mask 1, giving rise to a reconstruction of SNAPc-containing Pol II PIC bound to U5 promoter in CC state at 3.0-Å resolution (map 4) and a local map of the SNAPc-containing upstream complex extending to 3.2 Å (map 5).

The reported resolutions were calculated on the basis of the gold standard Fourier shell correlation (FSC) 0.143 criterion. After processing of the final reconstructions, B-factor sharpening was performed for all final maps on the basis of automatic B-factor determination in RELION (−5 Å$^2$ for map 1: SNAPc-PIC bound to U1 promoter in CC state, −10 Å$^2$ for map 2: SNAPc-PIC bound to U1 promoter in OC state and −10 Å$^2$ for map 3: local map of SNAPc-containing upstream complex, −10 Å$^2$ for map 4: SNAPc-PIC bound to U5 promoter in CC state and −10 Å$^2$ for map 5: local map of SNAPc-containing upstream complex). Estimates of local resolution were calculated using the in-built local-resolution tool of RELION and the estimated B-factors. To assist in model building, a local-resolution-filtered map (but unsharpened) of map 5 was sharpened locally using PHENIX.auto_sharpen[49].

## Model building and refinement
The PIC was modeled using the core PIC part of the previously published high-resolution structures in closed and open promoter states[24]. For SNAPc, the subunits SNAPC1 and SNAPC4 were built using partial homology models generated using TrRosetta[50]. The partial models were rigid-body fitted into the density using UCSF Chimera[51] and were manually extended and corrected using Coot[52] to fit the density. The subunit SNAPC3 was modeled entirely de novo using the experimental density in Coot. Ambiguous density corresponding to linker regions was not modeled. The model corresponding to the wing-2 region constituting parts of SNAPC1, SNAPC3 and SNAPC4 was modeled using AlphaFold[29]. The model for promoter DNA in CC and OC states was obtained using the high-resolution structures of human PIC as template where in the sequence register was mutated to fit the U1 and U5 respectively. The models were then subjected to iterative rounds of PHENIX real-space refinement followed by manual adjustment in coot to achieve final models with good stereochemistry as assessed by MolProbity[53]. Figures representing the 3D structures and maps were prepared using PyMOL, UCSF Chimera and UCSF ChimeraX.

## Cross-linking mass-spectrometry
To prepare a sample for performing cross-linking mass-spectrometry, a stable complex of SNAPc-containing Pol-II PIC bound to U5 promoter was isolated. An assay containing Pol II, TBP, TFIIA, TFIIB, TFIIF and SNAPc-FL was incubated in ratios explained above and was subjected to size-exclusion chromatography using Superose 6 increase 3.2/300 GL column (GE Healthcare) pre-equilibrated with buffer-x (25 mM Hepes pH 7.5, 100 mM NaCl, 5 mM MgCl2, 5% glycerol and 2 mM TCEP).

The peak fractions were then pooled and incubated with 1 mM of bissulfosuccinimidyl suberate (BS3) for 45 minutes at 4 °C. The cross-linking reaction was quenched using a cocktail of 10 mM aspartate and 30 mM lysine.

Cross-linked proteins were resuspended in 4 M urea/50 mM ammonium bicarbonate for 10 minutes at 25 °C and reduced for 30 minutes at RT with 10 mM dithiothreitol. Proteins were alkylated for 30 minutes at RT in the dark by adding iodoacetamide (IAA) to a final concentration of 55 mM. Sample was diluted to 1 M urea and digested for 30 minutes at 37 °C with 4 µl Pierce Universal Nuclease (250 U/µl) in the presence of 2 mM MgCl2. Trypsin (Promega) digest was performed overnight at 37 °C in a 1:50 enzyme/protein ratio, and the reaction was terminated with 0.2 % (vol/vol) FA. Tryptic peptides were desalted on MicroSpin Columns (Harvard Apparatus), following the manufacturer's instructions, and were vacuum-dried. Cross-linked peptides were resuspended in 50 µl 30% acetonitrile/0.1 % TFA and enriched by peptide size-exclusion chromatography/pSEC (Superdex Peptide PC3.2/300 column, GE Healthcare, flow rate 50 µl/min).

Cross-linked peptides derived from pSEC were subjected to liquid chromatography mass spectrometry (LC–MS) on a Thermo Orbitrap Exploris mass spectrometer. Peptides were loaded in duplicates onto a Dionex Ultimate 3000 RSLCnano equipped with a custom column (ReproSil-Pur 120 C18-AQ, 1.9 µm pore size, 75 µm inner diameter, 30 cm length, Dr. Maisch). Peptides were separated applying the following gradient: mobile phase A consisted of 0.1% formic acid (FA, vol/vol), mobile phase B of 80% ACN/0.08% FA (vol/vol). The gradient started at 5% B, increasing to 10%, 15% or 20% B within 3 minutes, followed by a continuous increase to 48% B within 45 minutes, then keeping B constant at 90% for 8 minutes. After each gradient, the column was again equilibrated to 5% B for 2 minutes. The flow rate was set to 300 nl/min.

MS1 spectra were acquired with a resolution of 120,000 in the orbitrap (OT) covering a mass range of 380–1600 $m/z$. Dynamic exclusion was set to 30 seconds. Only precursors with a charge state of 3-8 were included. MS2 spectra were recorded with a resolution of 30,000 in OT and the isolation window set to 1.6 $m/z$. Fragmentation was enforced by higher-energy collisional dissociation (HCD) at 30%. Raw files were searched against a database containing the sequences of the proteins of the complex and analyzed via pLink 2.3.9 at a false discovery rate (FDR) of 1% (ref. [54]). Carbamidomethylation of cysteines was set as fixed modification and oxidation of methionines as variable modification. The database contained all proteins within the complex. For further analysis, only interaction sites with three cross-linked peptide-spectrum matches were taken into account. Cross-links were displayed with xiNET and XlinkAnalyzer in UCSF Chimera.[51,55,56]

## TSS precision analyzes in cells

We used published 5′ cap-seq data[36] (GEO: GSE159633) for analyzes of TSS precision in cells. The raw data were processed as described previously[36] to obtain the 5′ ends of reads and to generate normalized coverage. In brief, we first removed the unique molecular identifier (UMI) from 5′ cap-seq reads with UMI-tools[57] and then trimmed adapter sequences with Cutadapt[58] and mapped them to the human genome (GRCh38) merged with the *D. melanogaster* genome (Dm6) with the STAR mapper[59]. We next deduplicated the mapped data with UMI-tools to remove any PCR duplicates and then determined the first transcribed base and used this position in downstream analyzes. Normalization factors were obtained from the spike-in reads (processed as above) that mapped to the spike-in genome and were used to normalize the human genome coverage profiles. The replicates were combined by summing the normalized coverage per nucleotide. Thus, obtaining genome-wide capped 5′ end signal (5′ cap-seq signal) at single-base resolution. We subset the NCBI reference genome annotation[60] (GRCh38.p7) to contain only genes annotated to the primary assembly and to include only genes with known transcripts (prefix: 'NR' or 'NM') and to also exclude overlapping genes. To exclude genes with alternative start sites from downstream analyzes, we included only genes that have a constitutive first or a single exon in our downstream analyzes.

To determine the main TSS, we determined the position with the highest 5′ cap-seq signal within constitutive first exons of the reference annotation. To accommodate for reference annotation imprecision, we also included 10 bp upstream of the annotated TSS and set the downstream cut-off to 500 bp downstream of the annotated TSS. We thus obtained the main TSS for each constitutive TSS. We next quantified the 5′ cap-seq signal of the main TSS (±2 bp) and the TSS region (main TSS ± 50 bp). We excluded genes with fewer than 10 counts in the TSS region and genes with biotypes that are not either protein-coding or snRNA. From the remaining annotated snRNA subset, we also removed known Pol III transcripts: RN7SK, RNU6ATAC, SNAR-G2, RNU6-2, SNAR-C4, SNAR-G1 and SNAR-C3, and identified protein-coding gene promoters that contain a TATA-box motif (JASPAR database, 2020 release: https://jaspar2020.genereg.net/matrix/POL012.1/) within 50 bp upstream of the annotated TSS. Finally, we determined the TSS precision score by dividing the TSS peak counts by the TSS region counts. The maximum TSS precision score is 1, which means that all 5′ cap-seq signal is within the TSS peak. The preprocessed 5′ cap-seq data were analyzed in RStudio[61] with R version 3.6.1 (ref. [62]) and packages from the Bioconductor repository[63,64] and Tidyverse[65]. Plots were generated with ggplot2 and ggbio[66].

## Statistics and reproducibility

No statistical method was used to predetermine sample size. No data were excluded from the analyses. The experiments were not randomized. The investigators were not blinded to allocation during experiments and outcome assessment.

## Reporting summary

Further information on research design is available in the Nature Research Reporting Summary linked to this article.

## Data availability

The cryo-EM density reconstructions were deposited to the EMDB under accession codes EMD-14996 (U5-CC), -14997 (U5-local), -15006 (U1-CC), -15007(U1-OC), -15009(U1-local) and the respective atomic coordinates were deposited to the PDB under the accession codes PDB-7ZWC, -7ZWD, -7ZX7, -7ZX8, -7ZXE. The mass spectrometry proteomics data have been deposited to the ProteomeXchange Consortium via the PRIDE[67] partner repository with the dataset identifier PXD033638. All data is available in the main text or the supplementary materials. Source data are provided with this paper.

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

## Acknowledgements

We thank present and past members of the Cramer laboratory for help and discussions. We thank F. Grabbe for purifying TBP, TFIIA, TFIIB, TFIIE, TFIIF and TFIIH. We thank C. Dienemann and S. Schilbach for input towards designing the engineered U5 promoter DNA. We thank C. Diederich for discussions about AlphaFold2. We thank C. Dienemann and U. Steuerwald for maintenance of the electron microscopy facility. S. R. was supported by a postdoctoral fellowship from Peter und Traudl Engelhorn Foundation. A. V. was supported by the Cancer Research UK Programme Foundation (CR-UK C47547/A21536) and a Wellcome Trust Investigator Award (200818/Z/16/Z). H. U. was supported by the Deutsche Forschungsgemeinschaft (SFB860). P. C. was supported by the Deutsche Forschungsgemeinschaft (EXC 2067/1 39072994, SFB860) and the ERC Advanced Investigator Grant CHROMATRANS (grant agreement No. 882357).

## Author contributions

S. R. carried out all experiments and data analysis, unless stated otherwise. S. S. performed the in vitro transcription assay and quantification. T. K. and J. G. cloned, expressed and purified the SNAPc variants and performed EMSA assays. K. Z. performed the reanalysis of 5' cap-seq data, TSS precision plots and the web-logo plots. J. S. performed cross-linking mass-spectrometry and data analysis and was supervised by H. U. S. R. and C. D. collected the cryo-EM datasets. P. C. and A. V. designed and supervised research. S. R. and P. C. interpreted the data and wrote the manuscript, with input from all authors.

## Funding

## Competing interests

The authors declare no competing interests.

## Additional information

**Extended data** is available for this paper at https://doi.org/10.1038/s41594-022-00857-w.

**Correspondence and requests for materials** should be addressed to Alessandro Vannini or Patrick Cramer.

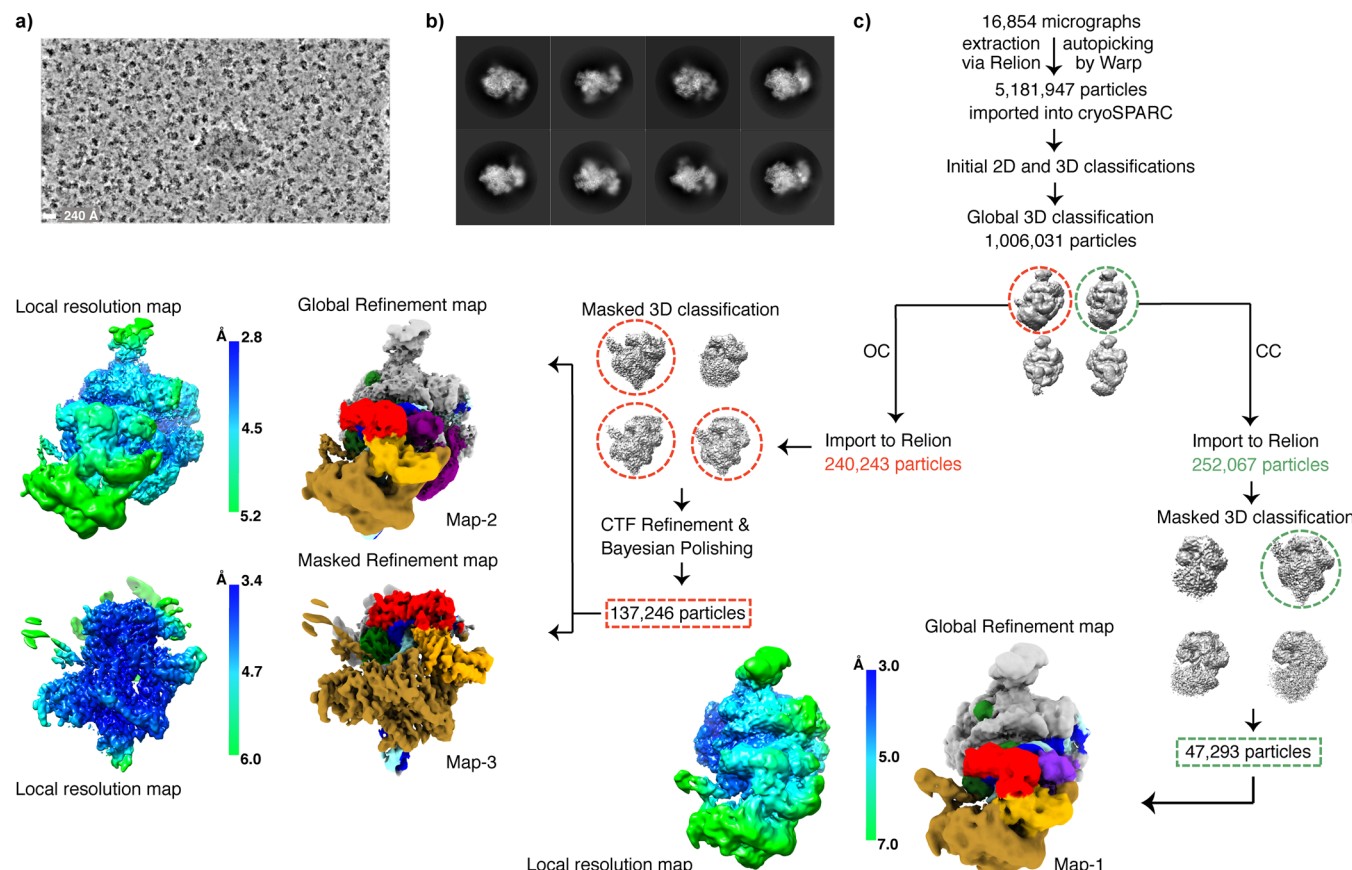

**Extended Data Fig. 1 | Processing of cryo-EM data for SNAPc-containing Pol II PIC bound to U1 promoter. Related to Fig. 2. a)** Representative cryo-EM micrograph (out of 16, 854 in total) of the SNAPc-containing Pol II PIC bound to U1 promoter cryo-EM data collection. Scale bar – 240 Å. **b)** Representative 2D class averages of initially sorted datasets after merging. Adjacent to a well-defined PIC, clear signal for SNAPc is detected. **c)** Complete processing scheme. After initial clean-up procedures, particles representing SNAPc containing PIC were recovered as two sets. These particle sets were processed separately with respect to the promoter DNA state (CC/OC) and SNAPc occupancy. Final maps are coloured using the subunit color code in Fig. 1. The local resolution map indicate the resolution range of final maps (scale bar).

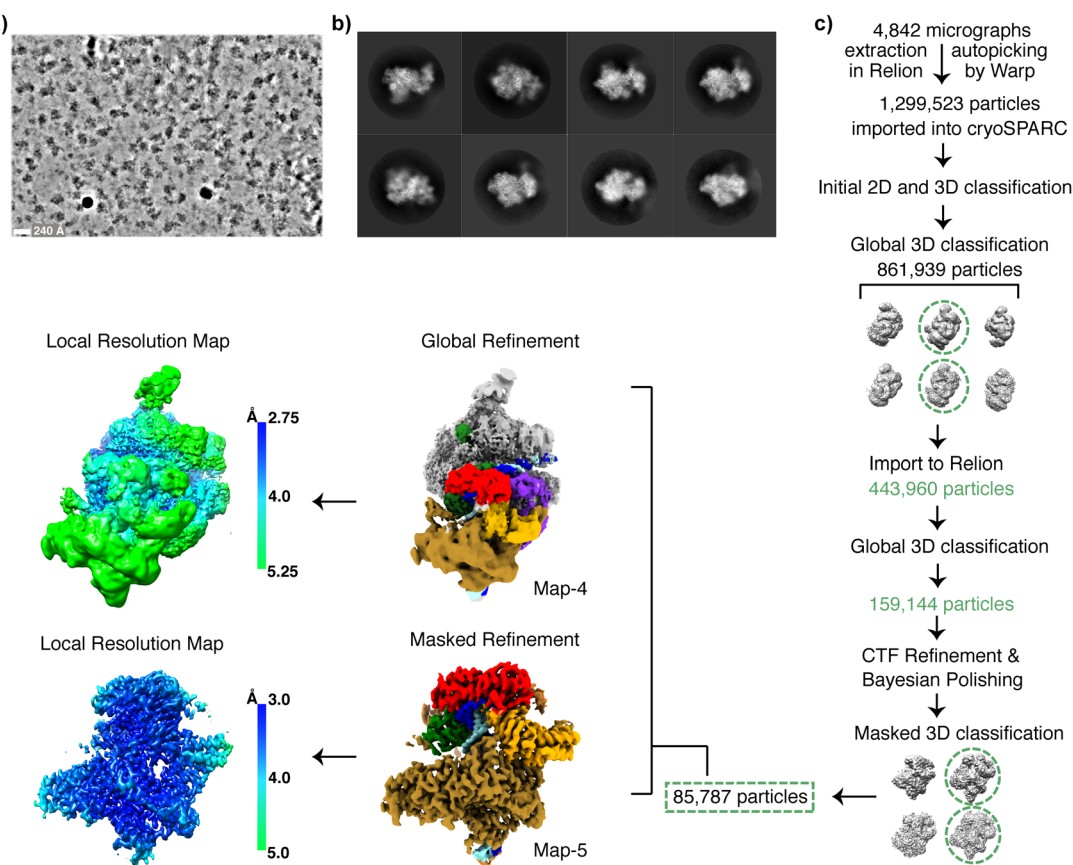

**Extended Data Fig. 2 | Processing of cryo-EM data for SNAPc-containing Pol II PIC bound to U5 promoter. Related to Fig. 2. a**) Representative cryo-EM micrograph (out of 4842 in total) of the SNAPc-containing Pol II PIC bound to U5 promoter cryo-EM data collection. Scale bar – 240 Å. **b**) Representative 2D class averages of initially sorted datasets after merging. As in the case of U1 promoter dataset, a clear signal for SNAPc is detected adjacent to a well-defined PIC. **c**) Complete processing scheme. The optimized strategy from U1 promoter bound SNAPc-PIC dataset was used to obtain high resolution maps of SNAPc-PIC bound to U5 promoter. Final maps are coloured using the subunit color code in Fig. 1. The local resolution map of global and locally refined maps indicate the resolution range of final maps (scale bar).

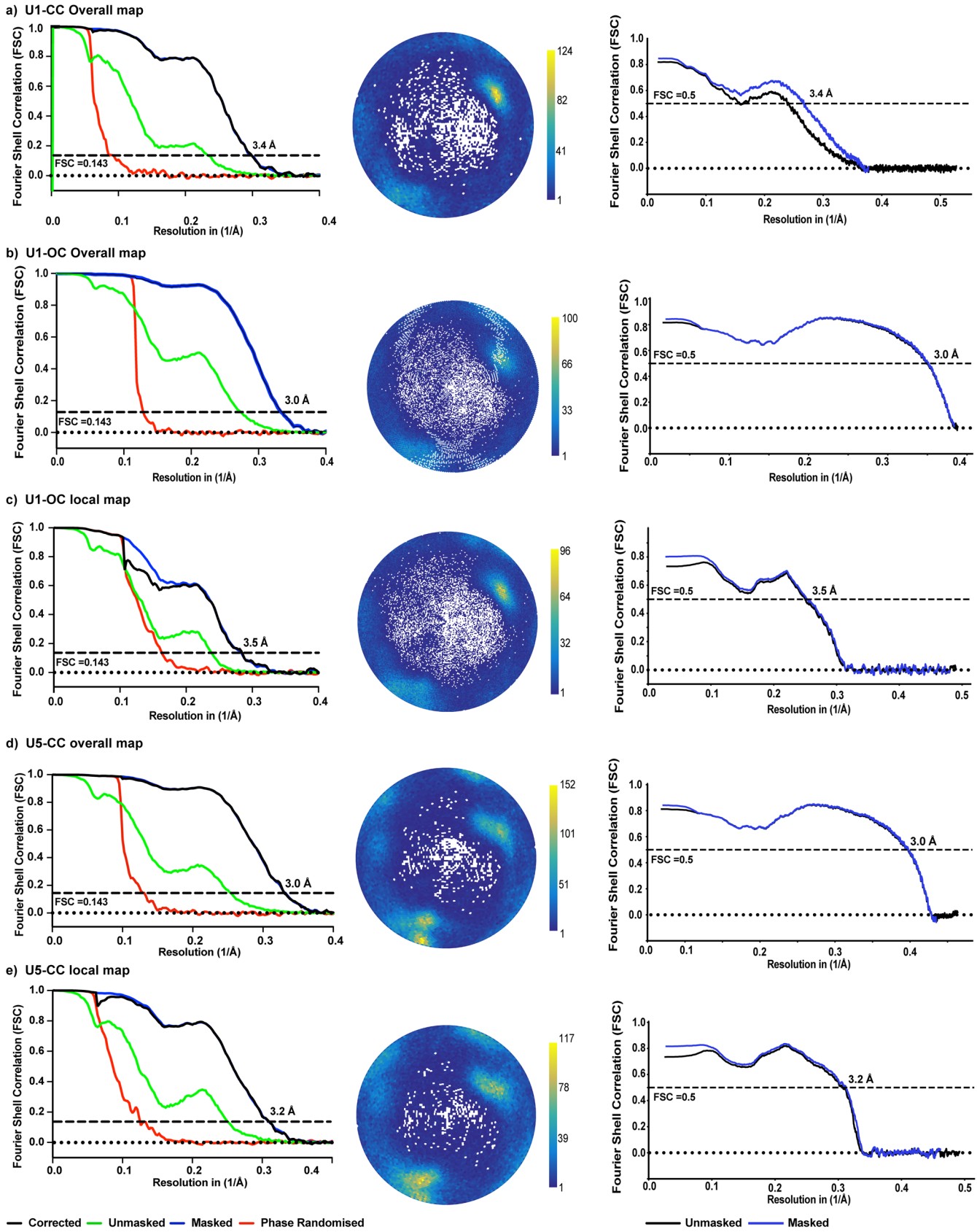

**Extended Data Fig. 3 | FSC and angular distribution plot of cryo-EM reconstructions. Related to Fig. 2. a-e)** On the left - FSC plot showing the overall resolution of the reconstructions determined by the gold standard FSC cut-off 0.143, indicated in the graph. In the middle – angular distribution plot of the respective reconstruction showing assignment of particles with respect to various angles. Colour bar indicates number of samples per angular bin (white areas indicate unpopulated angles). On the right - Model-to-map FSCs, showing the fit of modelled structures to their corresponding maps.

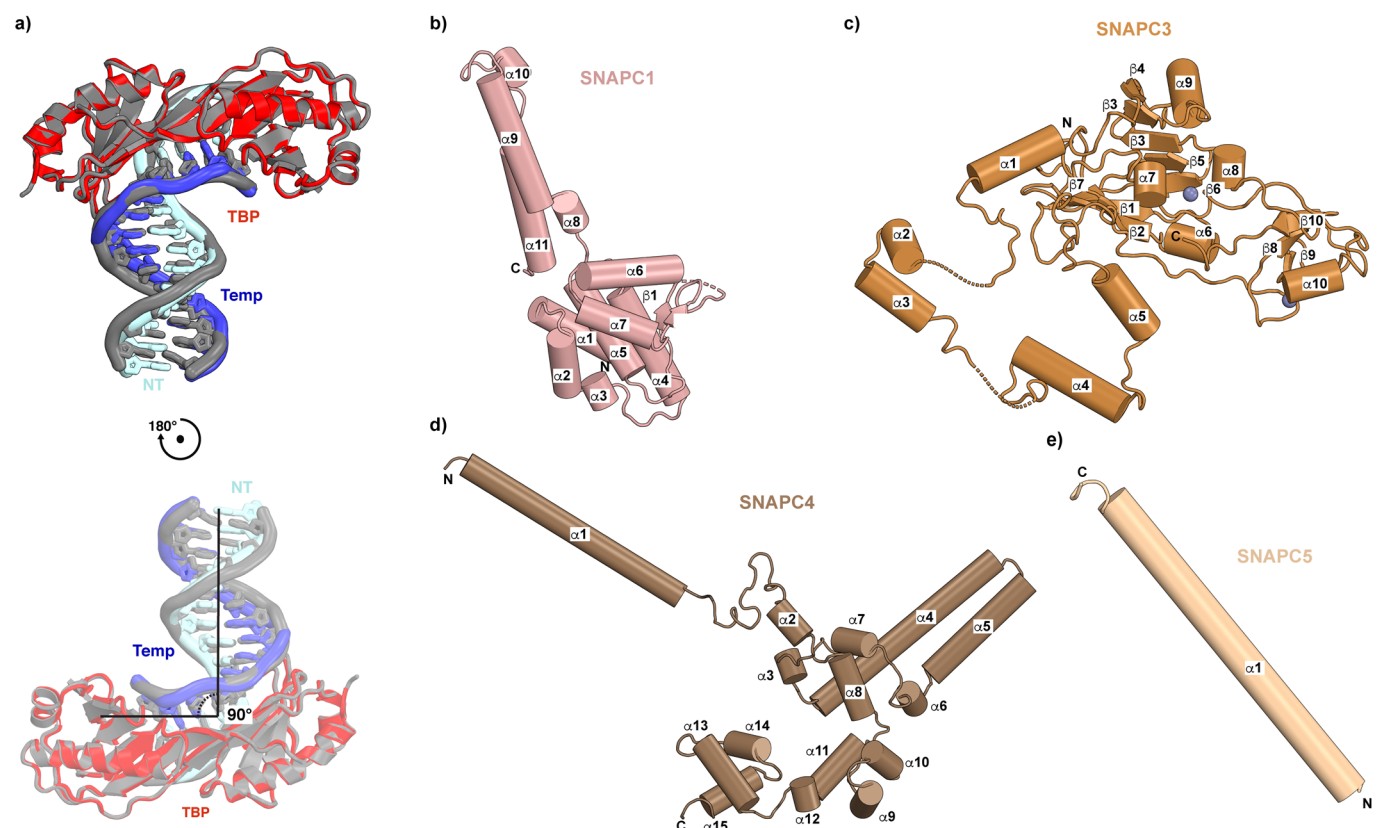

**Extended Data Fig. 4 | Structural comparison of TBP bound to TATA containing and TATA-less DNA template; Overall of Structure of individual SNAPc subunits. Related to Figs. 2 and 3. a)** Structural super-position of TBP(red) bound TATA-less U1 promoter (cyan/blue) on to TBP (grey) bound to TATA box sequence (PDB: 1YTF)(Tan et al.,[26]). The comparison shows that TBP binds to the TATA-less sequence in a canonical fashion and bends the DNA by 90°. **b-e)** Cartoon representation of the individual structures of SNAPc subunits SNAPC1, 3, 4 and 5 displaying its secondary structure elements as labelled. The N and C termini of all subunits are indicated.

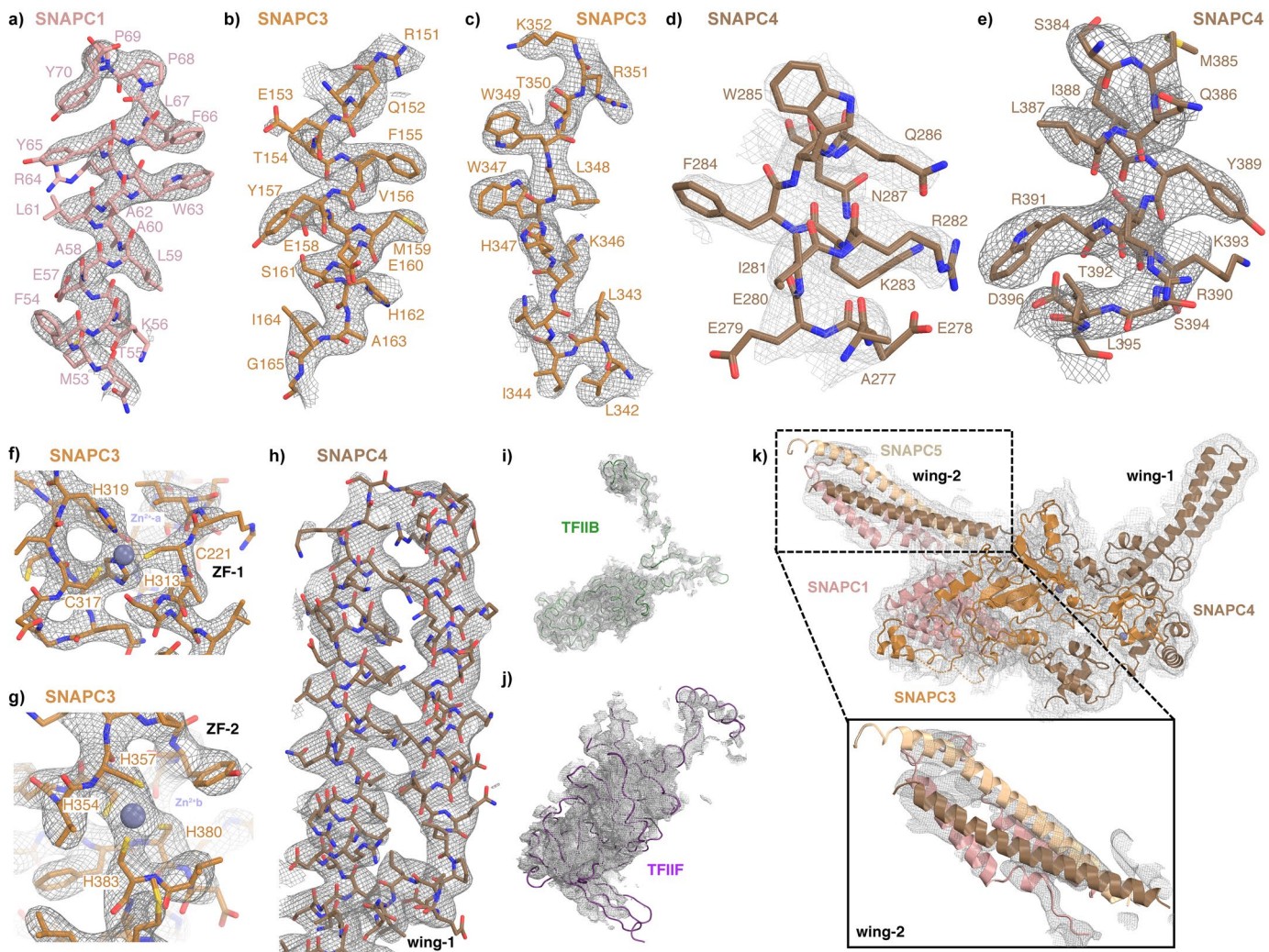

**Extended Data Fig. 5 | Map quality and map to model fit. Related to Fig. 3.**
**a-h**) Sections of cryo-EM density of SNAPc subunits overlaid with their respective atomic models. Densities are shown as a grey mesh, and sticks are shown for the model as coloured in Fig. 3. **i**) cryo-EM density of the TFIIB subunit overlaid to the atomic within the SNAPc containing Pol II PIC bound to U1 promoter in OC state.

**j**) cryo-EM density of a region of TFIIF subunit overlaid to the atomic model within the SNAPc containing Pol II PIC bound to U1 promoter in OC state. **d**) Local map of SNAPc containing Pol II PIC bound to U1 promoter in OC state is low pass filtered to 5 Å. The corresponding map is fitted with SNAPc subunits representing map to model fit, in particular the 'wing-2' region modelled using AlphaFold2[1].

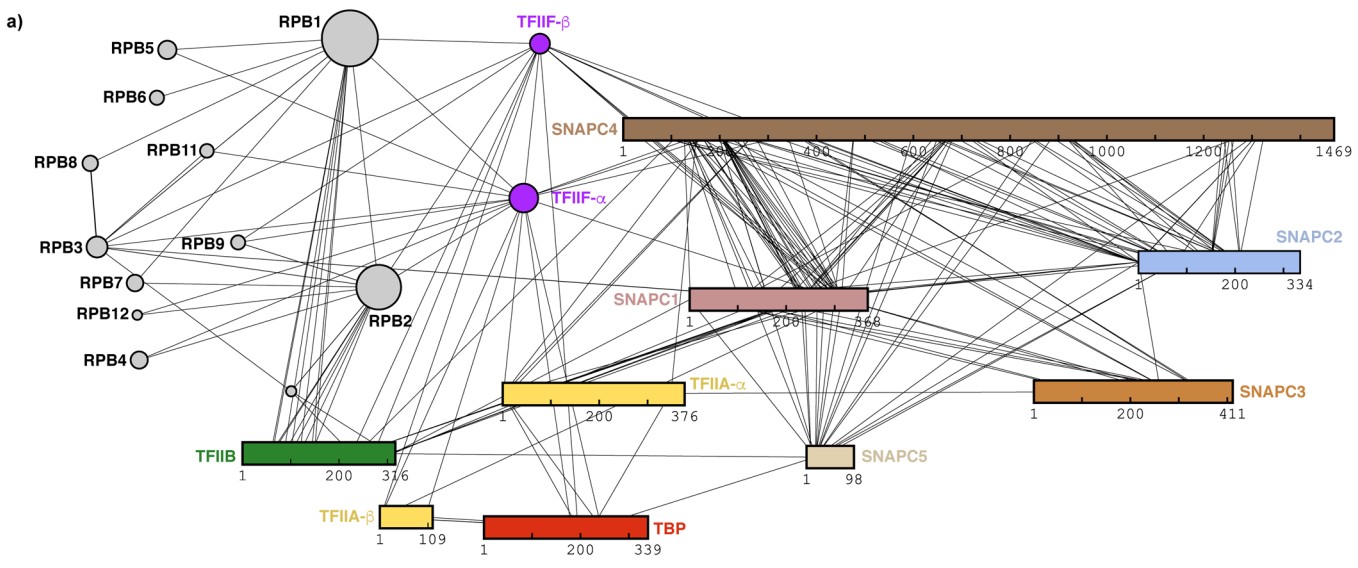

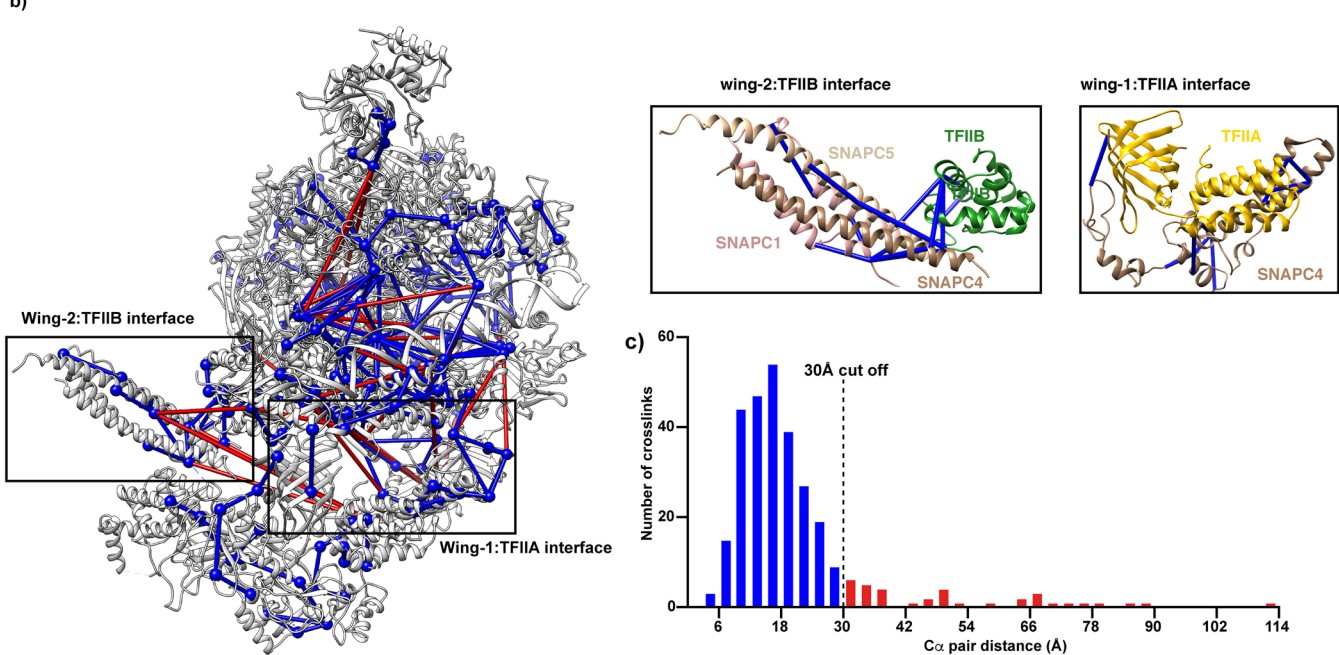

**Extended Data Fig. 6 | Crosslinking mass-spectrometric analysis of SNAPc containing Pol II PIC. Related to Figs. 2 and 3. a)** 2D representation of the overview of BS3 crosslinks. The crosslinks correspond to inter-protein mono-links that have at least three crosslinked peptide-spectrum matches (CSM). The subunit colours are consistent with Fig. 2. **b)** Crosslinks as mapped to SNAPc containing Pol II PIC structure using Xlink analyzer[2] plugin in UCSF chimera. The inset show the crosslinks observed between SNAPc subunits and the GTFs' TFIIA and TFIIB respectively. **c)** Histogram representing the distribution of Cα pair distances of unique crosslinks mapped to the structure. Dotted line indicates the 30 Å cut-off for BS3 crosslinked Cα pair. A total of 87.8% of the crosslinks were satisfied within this 30 Å cutoff.

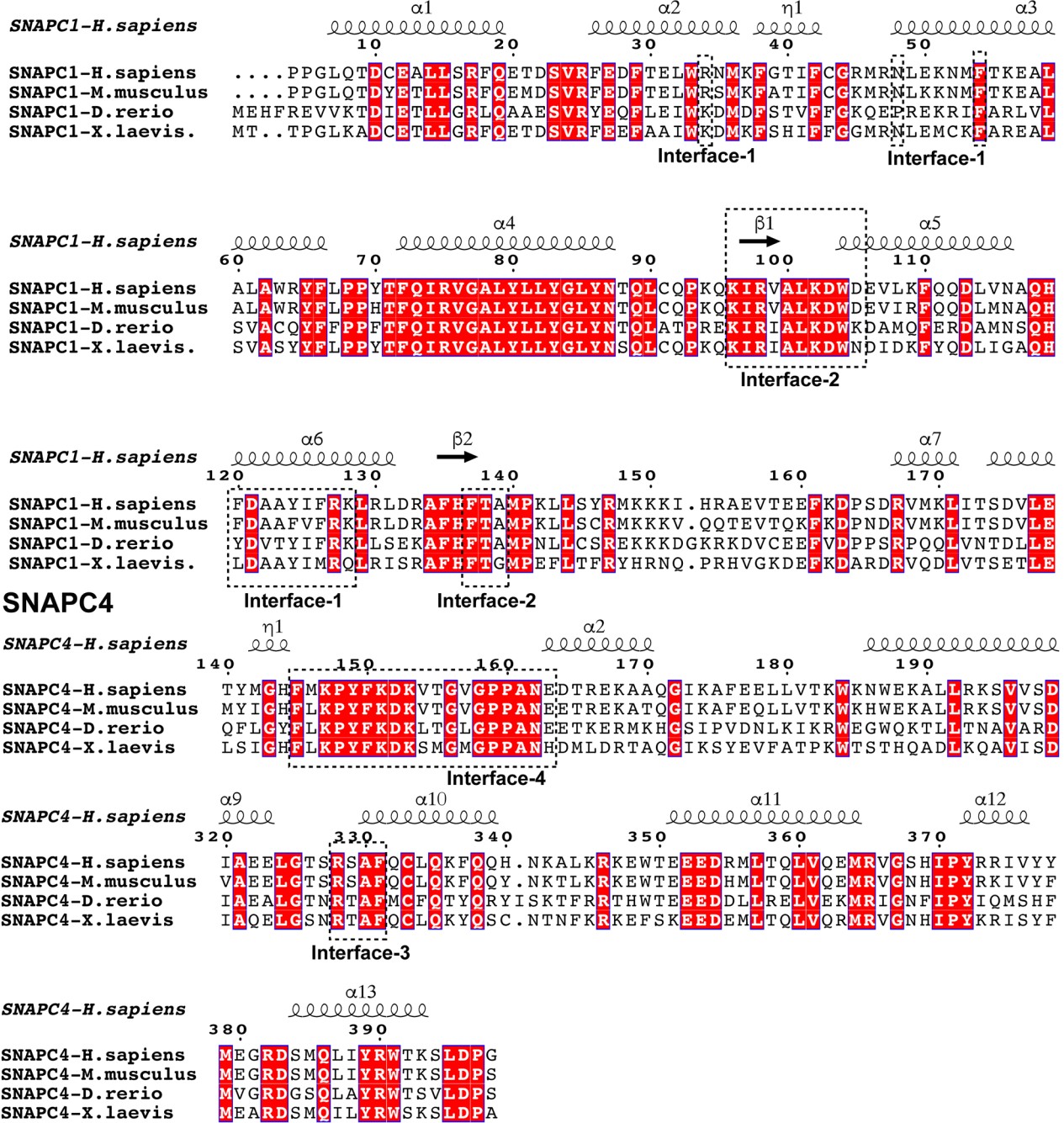

**Extended Data Fig. 7 | Structure based sequence alignment of the subunits SNAPC1 and SNAPC4 involved in interactions. Related to Figs. 3 and 4.** Sequence alignments were performed with the regions of individual subunits for which the structure has been determined in this study. T-Coffee algorithm[3] was adopted to obtain a structure based sequence alignment which was then visualized using ESPript[4]. Residues with identity above 80% are coloured red. Regions involved in interactions are indicated by dashed boxes and labels.

## SNAPC3

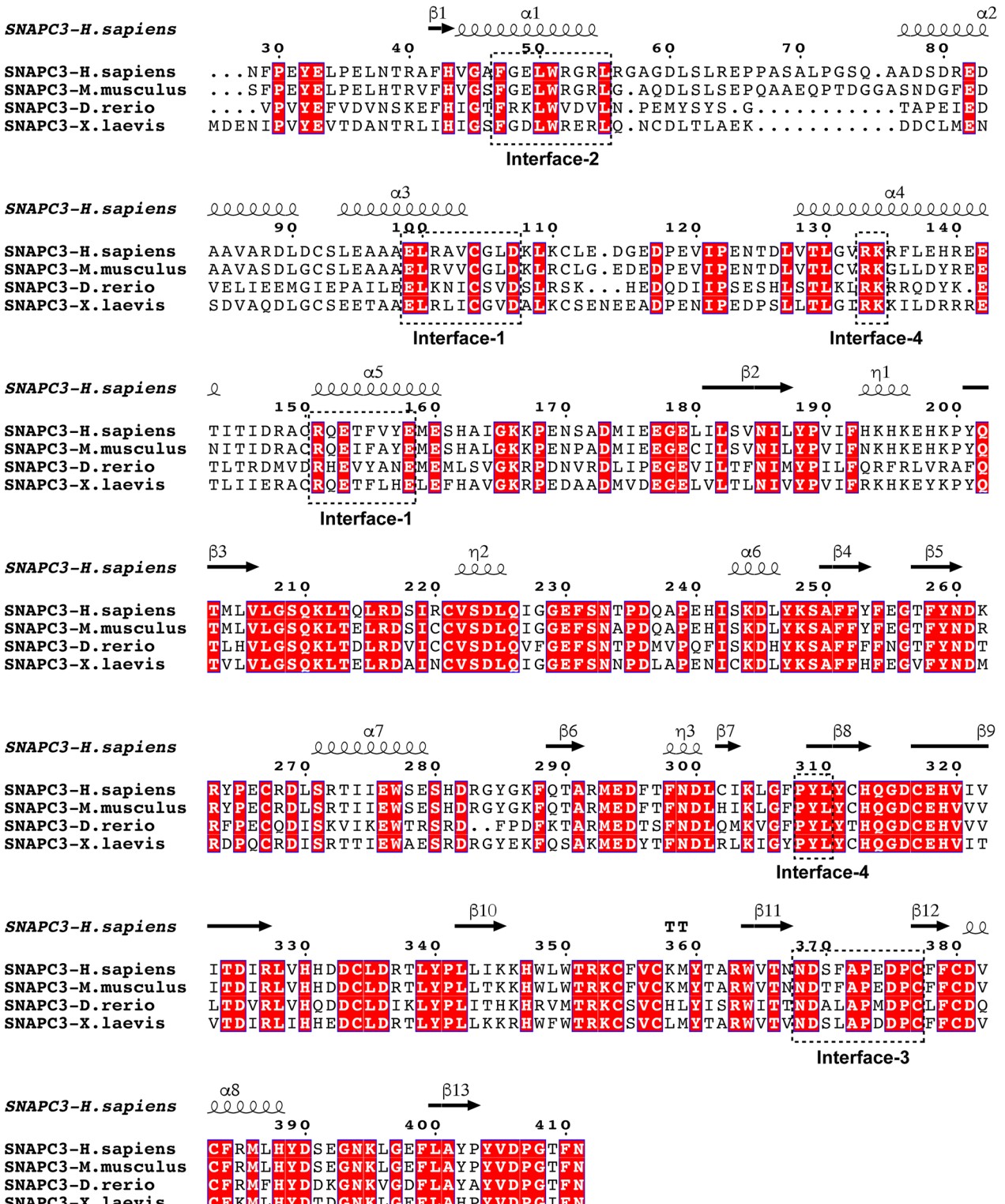

**Extended Data Fig. 8 | Structure based sequence alignment of the subunit SNAPC3 involved in interactions. Related to Figs. 3 and 4.** Sequence alignment was performed as described in Extended Data Fig. 7. Residues with identity above 80% are coloured red. Regions involved in interactions are indicated by dashed boxes and labels.

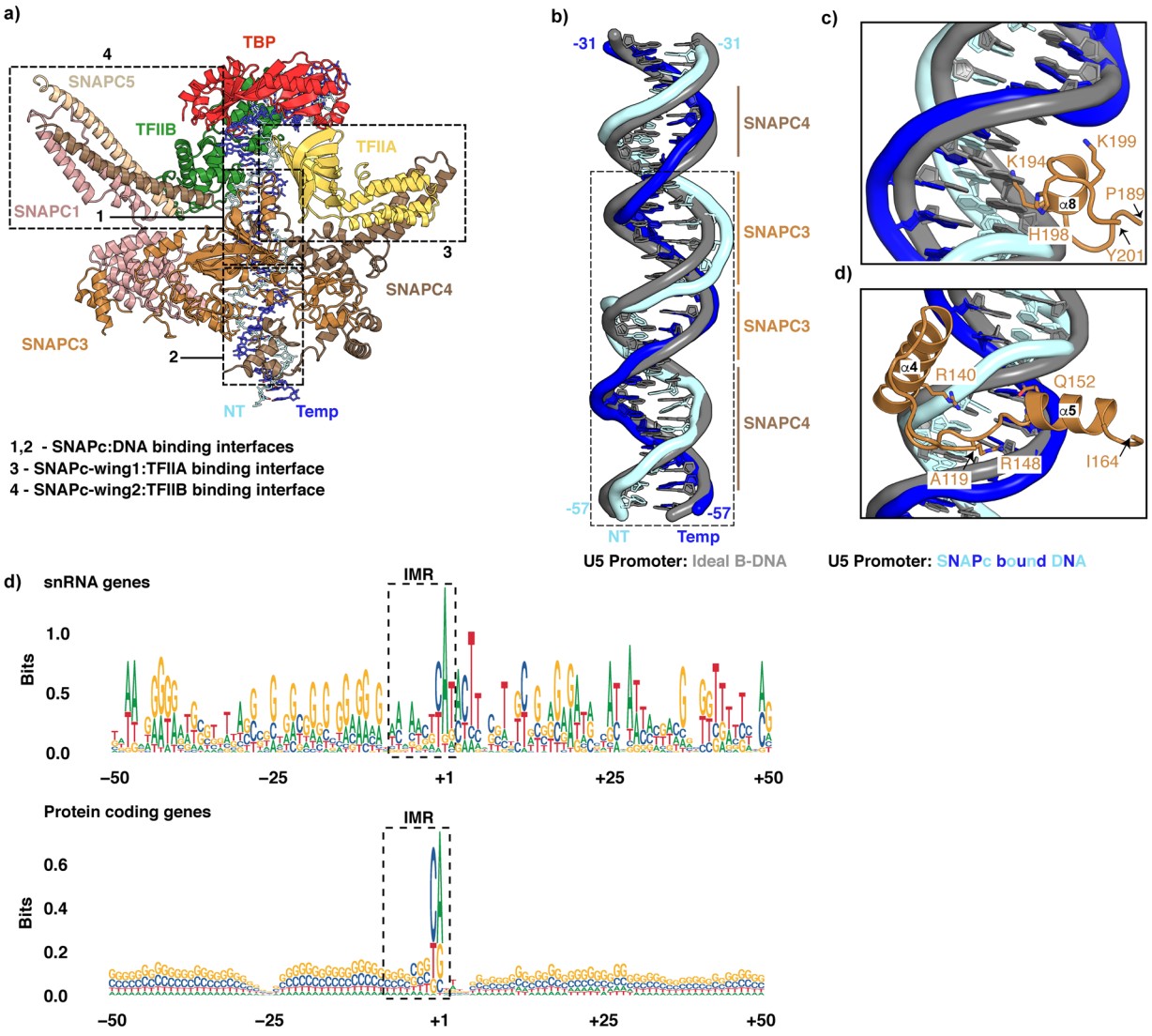

**Extended Data Fig. 9 | Related to Figs. 4, 5 and 6.** Extended Data Fig. 8 **a**) Birds-eye view of the SNAPc interaction with the GTFs' and the PSE motif on U5 snRNA promoter. The dashed boxes indicate the observed interaction surfaces within the complex (1–4). **b**) Structural super-position of ideal B-DNA of U5 promoter to the SNAPc bound experimental DNA structure. Major and minor grooves of U5 promoter bound by SNAPC3 and SNAPC4 are labelled and highlighted with lines. Dashed box indicates the PSE region. **c**) Close up view of SNAPC3 helix α8 binding to major groove of U5 promoter. The observed steric clash of K194 with B-DNA highlights the distortion upon SNAPc binding. **d**) Close up view of SNAPC3 helices α4, α5 region binding to minor groove of U5 promoter. The views in panels c and d correspond to Fig. 4b, c. **e**) Sequence logos of DNA sequence surrounding TSS peaks in expressed constitutive first/single exons for all snRNA genes (n = 18) and protein coding genes (n = 4721), sorted by TSS precision scores. The boxes indicate the IMR region (−8 to +2) of promoter flanking the TSS (+1). While the protein coding genes do not show any enrichment of specific nucleotides, snRNA genes present a AT-rich profile in the IMR region, indicating its tendency for spontaneous promoter opening.

# Reporting Summary

Nature Research wishes to improve the reproducibility of the work that we publish. This form provides structure for consistency and transparency in reporting. For further information on Nature Research policies, see our Editorial Policies and the Editorial Policy Checklist.

## Statistics

For all statistical analyses, confirm that the following items are present in the figure legend, table legend, main text, or Methods section.

| n/a | Confirmed | |
|---|---|---|
| ☐ | ☒ | The exact sample size (*n*) for each experimental group/condition, given as a discrete number and unit of measurement |
| ☐ | ☒ | A statement on whether measurements were taken from distinct samples or whether the same sample was measured repeatedly |
| ☒ | ☐ | The statistical test(s) used AND whether they are one- or two-sided *Only common tests should be described solely by name; describe more complex techniques in the Methods section.* |
| ☒ | ☐ | A description of all covariates tested |
| ☒ | ☐ | A description of any assumptions or corrections, such as tests of normality and adjustment for multiple comparisons |
| ☐ | ☒ | A full description of the statistical parameters including central tendency (e.g. means) or other basic estimates (e.g. regression coefficient) AND variation (e.g. standard deviation) or associated estimates of uncertainty (e.g. confidence intervals) |
| ☒ | ☐ | For null hypothesis testing, the test statistic (e.g. *F*, *t*, *r*) with confidence intervals, effect sizes, degrees of freedom and *P* value noted *Give P values as exact values whenever suitable.* |
| ☒ | ☐ | For Bayesian analysis, information on the choice of priors and Markov chain Monte Carlo settings |
| ☒ | ☐ | For hierarchical and complex designs, identification of the appropriate level for tests and full reporting of outcomes |
| ☒ | ☐ | Estimates of effect sizes (e.g. Cohen's *d*, Pearson's *r*), indicating how they were calculated |

*Our web collection on statistics for biologists contains articles on many of the points above.*

## Software and code

Policy information about availability of computer code

| Data collection | Serial EM 3.8 beta 8 |
|---|---|
| Data analysis | RELION v3.1.0, UCSF Chimera 1.13, UCSF ChimeraX v1.11, Pymol 2.3.4, Coot 0.8.9.2, Warp v1.0.7-1.0.9, PHENIX 1.18.2, cryoSPARC 3.2.0, RStudio, R version 3.6.1, ggplot2, ggbio. |

For manuscripts utilizing custom algorithms or software that are central to the research but not yet described in published literature, software must be made available to editors and reviewers. We strongly encourage code deposition in a community repository (e.g. GitHub). See the Nature Research guidelines for submitting code & software for further information.

## Data

Policy information about availability of data

All manuscripts must include a data availability statement. This statement should provide the following information, where applicable:
- Accession codes, unique identifiers, or web links for publicly available datasets
- A list of figures that have associated raw data
- A description of any restrictions on data availability

The cryo-EM density reconstructions were deposited to the EMDB under accession codes EMD-14996 (U5-CC), -14997 (U5-local), -15006 (U1-CC), -15007(U1-OC), -15009(U1-local) and the respective atomic coordinates were deposited to the PDB under the accession codes PDB-7ZWC, -7ZWD, -7ZX7, -7ZX8, -7ZXE. The mass spectrometry proteomics data have been deposited to the ProteomeXchange Consortium via the PRIDE partner repository with the dataset identifier PXD033638. All data is available in the main text or the supplementary materials.

# Field-specific reporting

Please select the one below that is the best fit for your research. If you are not sure, read the appropriate sections before making your selection.

☒ Life sciences ☐ Behavioural & social sciences ☐ Ecological, evolutionary & environmental sciences

For a reference copy of the document with all sections, see nature.com/documents/nr-reporting-summary-flat.pdf

# Life sciences study design

All studies must disclose on these points even when the disclosure is negative.

| | |
|---|---|
| Sample size | No Sample size calculations were performed. For cryo-EM samples, at least eleven grids of SNAPc-PIC (U1/U5 promoter)sample were pre-screened to identify the optimal grid for data collection. The number of grids screened were random and was not limited by any experimental parameter.<br>Biochemical experiments were performed with three sample replicates and each experiment was repeated minimum thrice. This is a standard in the field and the sample size was sufficient to observe the effect and binary outcome of this experiment. i.e. SNAPc activation of Pol II PIC in the in vitro transcription assay |
| Data exclusions | No data were excluded from the analyses. |
| Replication | All attempts at replication were successful. Cryo-EM single particle analysis inherently relies on averaging over a large number of independent observations. During the processing pipeline, replicate reconstructions were calculated over 3 times during the polishing and other related refinement procedures, yielding the same results at different resolutions. The in vitro transcription assay and EMSA were repeated minimum thrice. |
| Randomization | Samples were not allocated to groups. |
| Blinding | Blinding is not applicable for this study, as group allocation is not used. |

# Reporting for specific materials, systems and methods

We require information from authors about some types of materials, experimental systems and methods used in many studies. Here, indicate whether each material, system or method listed is relevant to your study. If you are not sure if a list item applies to your research, read the appropriate section before selecting a response.

## Materials & experimental systems

| n/a | Involved in the study |
|---|---|
| ☒ ☐ | Antibodies |
| ☐ ☒ | Eukaryotic cell lines |
| ☒ ☐ | Palaeontology and archaeology |
| ☒ ☐ | Animals and other organisms |
| ☒ ☐ | Human research participants |
| ☒ ☐ | Clinical data |
| ☒ ☐ | Dual use research of concern |

## Methods

| n/a | Involved in the study |
|---|---|
| ☒ ☐ | ChIP-seq |
| ☒ ☐ | Flow cytometry |
| ☒ ☐ | MRI-based neuroimaging |

# Eukaryotic cell lines

Policy information about cell lines

| | |
|---|---|
| Cell line source(s) | Hi5 cells: Expression Systems, Tni Insect cells in ESF921 media, item 94-002F<br>Sf9 cells: ThermoFisher, Catalogue Number 12659017, Sf9 cells in Sf-9000TM III SFM |
| Authentication | None of the cell lines were authenticated. |
| Mycoplasma contamination | Cell lines were not tested for mycoplasma contamination. |
| Commonly misidentified lines<br>(See ICLAC register) | No commonly misidentified cell lines were used. |

