## [Peer Review File · Nature Structural & Molecular Biology]

Peer Review Information

Journal: Nature Structural and Molecular Biology

Manuscript Title: Structural basis of SNAPc-dependent snRNA transcription initiation by RNA polymerase II

Corresponding author name(s): Professor Patrick Cramer

Editorial Notes:

Reviewer Comments & Decisions:

Decision Letter, initial version:

Our ref: NSMB-A46118

29th Apr 2022

Dear Patrick,

Thank you again for submitting your manuscript "Structural basis of SNAPc-dependent snRNA transcription initiation by RNA polymerase II". The reports of the referees are below, and based on these comments, we are happy to accept your paper, in principle, for publication as an Article in Nature Structural & Molecular Biology, on the condition that you revise your manuscript in response to the comments of the referees and our editorial requirements.

I hope that you will be pleased to see that both reviewers are positive about the work, and that Reviewer #2 offers a particularly complementary summary of the insights provided by the findings. Each reviewer queries aspects of the in vitro transcription assays presented in Figure 1 that can be addressed by text and figure modifications. Reviewer #1 requests an expanded discussion of why TFIIE/TFIIH does not increase promoter-specific transcription in these assays and requests that a quantitative comparison of specific and non-specific transcripts be provided. Reviewer #2 similarly requests that the low in vitro stimulatory activity of SNAPc be noted and addressed in the text. Reviewer #1 also suggests including a panel showing SNAPc:IIA/IIB interactions in context of the full PIC. Editorially, we agree that these suggestions would strengthen the presentation of the findings and ask that they be included in a revised manuscript.

Note that, within a few days, we will send you detailed instructions for the final revision, along with information on editorial and formatting requirements. We recommend that you do not start revising the manuscript until you receive this additional information.

*****To facilitate our work at this stage, we would appreciate if you could send us the main text as a Word file. Please make sure to copy the NSMB account (cc'ed above).*****

TRANSPARENT PEER REVIEW

Nature Structural & Molecular Biology offers a transparent peer review option for new original research manuscripts submitted from 1st December 2019. We encourage increased transparency in peer review by publishing the reviewer comments, author rebuttal letters and editorial decision letters if the authors agree. Such peer review material is made available as a supplementary peer review file. **Please state in the cover letter 'I wish to participate in transparent peer review' if you want to opt in, or 'I do not wish to participate in transparent peer review' if you don't.** Failure to state your preference will result in delays in accepting your manuscript for publication.

If you have any questions, please do not hesitate to contact me directly.

With kind regards,

Beth

Beth Moorefield, Ph.D.
Senior Editor
Nature Structural & Molecular Biology

Reviewer #1 (Remarks to the Author):

Rengachari et al present several cryo-EM structures of SNAPc in complex with minimal Pol II preinitiation complex on U1 and U5 snRNA promoters. Using structural data complemented by in vitro transcription assays and EMSA the authors show that SNAPc in complex with minimal PIC allows for DNA melting in the absence of TFIIE and TFIIH. The authors also highlight the role of SNAPc extensions named wing-1 and wing-2 in binding TFIIA and TFIIIB, respectively, and coordinating SNAPc-PIC-promoter interaction. The authors compare open and closed promoter complexes and propose a mechanism of Pol II DNA opening in the absence of TFIIH.

Comments:

TFIIE density is not seen in structures, and this is supported by IVT experiments. However, after sucrose gradient ultracentrifugation it seems that both TFIIE subunits are present in complex with PIC-SNAPc in stoichiometric amounts (Fig 1 panel c). This needs more discussion.

Discuss why adding TFIIE/TFIIH does not increase transcription activity and leads to non-specific transcripts, even though their binding potentially is not restricted in the presented structures.

Extended Data Figure 5: typo, panel d) should be labeled as panel k)

Fig 6. Consider include a panel showing the interactions between SNAP and TFIIA/B in the context of the full complex.

Fig 1e. Show everything relative to PIC (w/o TFIIE/H). Quantitate the percentage of background products.

Ext Fig 6A. If you used a software to generate this figure please cite, otherwise disregard this comments.

In the methods please better define "scoop of DNase I"

Reviewer #2 (Remarks to the Author):

Rengachari et al have resolved cryo-EM structures of the SNAPc-containing Pol II preinitiation complex (PIC) assembled on U1 and U5 snRNA promoters. The structures show that the core of SNAPc binds two turns of DNA and recognizes the snRNA promoter-specific proximal sequence element (PSE) located upstream of the TATA box-binding protein TBP. Interestingly, SNAPc defines the TSS very precisely. The structures also show that two extensions of SNAPc called wing-1 and wing-2 bind TFIIA and TFIIB, respectively, explaining how SNAPc directs Pol II to snRNA promoters. Comparison of structures of closed and open promoter complexes indicate and explain that DNA opening is TFIIF-independent, unlike DNA opening for protein-coding gene initiation.

The results of this tour de force of structural and in vitro transcription analysis clearly provide the structural basis of Pol II initiation at PSE-containing promoters. As such, this is a timely and important piece of work.

I only have one quibble-in Figure 1d there is not a huge increase in transcription when SNAPc is added to the reaction, indicating that the in vitro conditions used are not great for PSE/SNAPc-dependent initiation. The authors should address why this might be.

Decision Letter, final checks:

Our ref: NSMB-A46118

16th May 2022

Dear Dr. Cramer,

Thank you for your patience as we've prepared the guidelines for final submission of your Nature Structural & Molecular Biology manuscript, "Structural basis of SNAPc-dependent snRNA transcription initiation by RNA polymerase II" (NSMB-A46118). Please carefully follow the step-by-step instructions provided in the attached file, and add a response in each row of the table to indicate the changes that you have made. Ensuring that each point is addressed will help to ensure that your revised manuscript can be swiftly handed over to our production team.

In recognition of the time and expertise our reviewers provide to Nature Structural & Molecular Biology's editorial process, we would like to formally acknowledge their contribution to the external peer review of your manuscript entitled "Structural basis of SNAPc-dependent snRNA transcription initiation by RNA polymerase II". For those reviewers who give their assent, we will be publishing their names alongside the published article.

Nature Structural & Molecular Biology offers a Transparent Peer Review option for new original research manuscripts submitted after December 1st, 2019. As part of this initiative, we encourage our authors to support increased transparency into the peer review process by agreeing to have the reviewer comments, author rebuttal letters, and editorial decision letters published as a Supplementary item. When you submit your final files please clearly state in your cover letter whether or not you would like to participate in this initiative. Please note that failure to state your preference will result in delays in accepting your manuscript for publication.

Cover suggestions

As you prepare your final files we encourage you to consider whether you have any images or illustrations that may be appropriate for use on the cover of Nature Structural & Molecular Biology.

Nature Structural & Molecular Biology has now transitioned to a unified Rights Collection system which will allow our Author Services team to quickly and easily collect the rights and permissions required to publish your work. Approximately 10 days after your paper is formally accepted, you will receive an

email in providing you with a link to complete the grant of rights. If your paper is eligible for Open Access, our Author Services team will also be in touch regarding any additional information that may be required to arrange payment for your article.

Please note that *Nature Structural & Molecular Biology* is a Transformative Journal (TJ). Authors may publish their research with us through the traditional subscription access route or make their paper immediately open access through payment of an article-processing charge (APC). Authors will not be required to make a final decision about access to their article until it has been accepted. [Find out more about Transformative Journals](https://www.springernature.com/gp/open-research/transformative-journals)

Authors may need to take specific actions to achieve [compliance with funder and institutional open access mandates](https://www.springernature.com/gp/open-research/funding/policy-compliance-faqs). If your research is supported by a funder that requires immediate open access (e.g. according to [Plan S principles](https://www.springernature.com/gp/open-research/plan-s-compliance)) then you should select the gold OA route, and we will direct you to the compliant route where possible. For authors selecting the subscription publication route, the journal's standard licensing terms will need to be accepted, including [self-archiving policies](https://www.springernature.com/gp/open-research/policies/journal-policies). Those licensing terms will supersede any other terms that the author or any third party may assert apply to any version of the manuscript.

Please use the following link for uploading these materials:
[Redacted]

Best regards,

Sophia Frank
Editorial Assistant
Nature Structural & Molecular Biology
nsmb@us.nature.com

On behalf of

Carolina Perdigoto, PhD
Chief Editor
Nature Structural & Molecular Biology
orcid.org/0000-0002-5783-7106

Reviewer #1:

Remarks to the Author:

Rengachari et al present several cryo-EM structures of SNAPc in complex with minimal Pol II preinitiation complex on U1 and U5 snRNA promoters. Using structural data complemented by in vitro transcription assays and EMSA the authors show that SNAPc in complex with minimal PIC allows for DNA melting in the absence of TFIIE and TFIIH. The authors also highlight the role of SNAPc extensions named wing-1 and wing-2 in binding TFIIA and TFIIB, respectively, and coordinating SNAPc-PIC-promoter interaction. The authors compare open and closed promoter complexes and propose a mechanism of Pol II DNA opening in the absence of TFIIH.

Comments:

TFIIE density is not seen in structures, and this is supported by IVT experiments. However, after sucrose gradient ultracentrifugation it seems that both TFIIE subunits are present in complex with PIC-SNAPc in stoichiometric amounts (Fig 1 panel c). This needs more discussion.

Discuss why adding TFIIE/TFIIH does not increase transcription activity and leads to non-specific transcripts, even though their binding potentially is not restricted in the presented structures.

Extended Data Figure 5: typo, panel d) should be labeled as panel k)

Fig 6. Consider include a panel showing the interactions between SNAP and TFIIA/B in the context of the full complex.

Fig 1e. Show everything relative to PIC (w/o TFIIE/H). Quantitate the percentage of background products.

Ext Fig 6A. If you used a software to generate this figure please cite, otherwise disregard this comments.

In the methods please better define "scoop of DNase I"

Reviewer #2:

Remarks to the Author:

Rengachari et al have resolved cryo-EM structures of the SNAPc-containing Pol II preinitiation complex (PIC) assembled on U1 and U5 snRNA promoters. The structures show that the core of SNAPc binds two turns of DNA and recognizes the snRNA promoter-specific proximal sequence element (PSE) located upstream of the TATA box-binding protein TBP. Interestingly, SNAPc defines the TSS very precisely. The structures also show that two extensions of SNAPc called wing-1 and wing-2 bind TFIIA and TFIIB, respectively, explaining how SNAPc directs Pol II to snRNA promoters. Comparison of

structures of closed and open promoter complexes indicate and explain that DNA opening is TFIID-independent, unlike DNA opening for protein-coding gene initiation.

The results of this tour de force of structural and in vitro transcription analysis clearly provide the structural basis of Pol II initiation at PSE-containing promoters. As such, this is a timely and important piece of work.

I only have one quibble—in Figure 1d there is not a huge increase in transcription when SNAPc is added to the reaction, indicating that the in vitro conditions used are not great for PSE/SNAPc-dependent initiation. The authors should address why this might be.

Author Rebuttal, First Revision:

Detailed list of responses to reviewer comments

NSMB-A46118

Responses are in blue italics

Referees' comments:

Reviewer #1 (Remarks to the Author):

Rengachari et al present several cryo-EM structures of SNAPc in complex with minimal Pol II preinitiation complex on U1 and U5 snRNA promoters. Using structural data complemented by in vitro transcription assays and EMSA the authors show that SNAPc in complex with minimal PIC allows for DNA melting in the absence of TFIIE and TFIID. The authors also highlight the role of SNAPc extensions named wing-1 and wing-2 in binding TFIIA and TFIIB, respectively, and coordinating SNAPc-PIC-promoter interaction. The authors compare open and closed promoter complexes and propose a mechanism of Pol II DNA opening in the absence of TFIID.

We thank the reviewer for their work. We have addressed the minor issues as described below.

TFIIE density is not seen in structures, and this is supported by IVT experiments. However, after sucrose gradient ultracentrifugation it seems that both TFIIE subunits are present in complex with PIC-SNAPc in stoichiometric amounts (Fig 1 panel c). This needs more discussion.

The approximate molecular weight of the Pol II PIC containing either SNAPc or TFIIE are similar. The resolution of the sucrose gradient is not high enough to distinguish between these two variants of the PIC and hence they co-migrate and are collected in the same gradient fractions. We observed this very early in our cryoEM data processing steps, where the PIC containing SNAPc or TFIIE separate as distinct sets of particles. The revised version of the manuscript includes a statement explaining this better.

Discuss why adding TFIIE/TFIIH does not increase transcription activity and leads to non-specific transcripts, even though their binding potentially is not restricted in the presented structures.

The exacerbated turnover of non-specific products evident for the PIC assembly containing both TFIIE and TFIIH are also mildly observed across the entire IVT experiment. This suggests the presence of a few putative non-specific binding events and start sites for Pol II PIC in the IVT template. In the absence of TFIIE and TFIIH, promoter opening and snRNA transcription occurs preferentially from the original PSE-directed start site. Once TFIIE and TFIIH are present, they appear to over-ride this start site preference because the sophisticated translocase machinery of TFIIH can enable opening of all the ungainly start sites. To a lesser extent, TFIIE alone is able to generate a similar effect, as has been observed in the yeast system (Plaschka, Hantse et al 2016, Dienemann et al 2019), on promoters with high meltability. A statement addressing this has been included in the updated version of the manuscript. With respect to the second point, TFIIE and TFIIH are not occluded sterically.

Extended Data Figure 5: typo, panel d) should be labeled as panel k)

The updated Extended Data Figure 5 has the suggested change incorporated.

Fig 6. Consider include a panel showing the interactions between SNAP and TFIIA/B in the context of the full complex.

This fine suggestion can help the reader with the bigger picture. SNAPc-TFIIA/TFIIB interaction belongs to Figure 5 and it has been updated accordingly.

Fig 1e. Show everything relative to PIC (w/o TFIIH). Quantitate the percentage of background products.

To address the reviewer's concern, we have prepared a version of Fig. 1e in which all bars are shown relative to PIC (w/o TFIIH) (see below). However, we refrain from the use of this version as it may confuse the reader and distract from the original message of the figure, which is to allude to the stimulatory effect of SNAPc on transcription.

We have quantified the background products and present the ratio between the main transcript and the background products in another separate figure (see below). There are two major observations. First, background transcription (all transcription that does not result in production of the main transcript) is increased by 4-7-fold in the presence of TFIIH. This points to the effectiveness of TFIIH in inducing DNA opening and transcription initiation also at such promoter sites as discussed above. Second, for the samples without TFIIH, background transcription is decreased for SNAPc-containing samples. This reflects

the ability of SNAPc to direct the Pol II machinery to the preferred TSS within the promoter sequence and assist in DNA opening at that position as discussed in the manuscript.

However, we would like to note that due to the nature of the experiment, which relies on quantification of bands from different urea-PAGE gels, the assignment of a defined baseline can vary between gels. Whereas this has no significant effect on quantification of a defined single band in the gel (such as the main transcript), this effect is much stronger for quantification of the background signal, which is the sum of all additional signal from a respective lane in the gel. This inconsistency is reflected in larger error bars. In order not to confuse the reader we would suggest not to include the background quantification figure in the manuscript. We also believe that it does not add any additional insights that are not obvious from the gel image itself. To this end, we have included a statement mentioning the 4-7-fold increased background signal in the presence of TFIIH in the legend for Figure 1e.

Ext Fig 6A. If you used a software to generate this figure please cite, otherwise disregard this comments.

We have cited all the software used for the generation of Ext Fig 6 in the methods section.

In the methods please better define "scoop of DNase I"

We have changed the text in the methods.

Reviewer #2 (Remarks to the Author):

Rengachari et al have resolved cryo-EM structures of the SNAPc-containing Pol II preinitiation complex (PIC) assembled on U1 and U5 snRNA promoters. The structures show that the core of SNAPc binds two turns of DNA and recognizes the snRNA promoter-specific proximal sequence element (PSE) located upstream of the TATA box-binding protein TBP. Interestingly, SNAPc defines the TSS very precisely. The structures also show that two extensions of SNAPc called wing-1 and wing-2 bind TFIIA and TFIIB, respectively, explaining how SNAPc directs Pol II to snRNA promoters. Comparison of structures of closed and open promoter complexes indicate and explain that DNA opening is TFIIH-independent, unlike DNA opening for protein-coding gene initiation.

The results of this tour de force of structural and in vitro transcription analysis clearly provide the structural basis of Pol II initiation at PSE-containing promoters. As such, this is a timely and important piece of work.

We thank the reviewer for a very positive assessment of our work.

I only have one quibble-in Figure 1d there is not a huge increase in transcription when SNAPc is added to the reaction, indicating that the in vitro conditions used are not great for PSE/SNAPc-dependent initiation. The authors should address why this might be.

Although SNAPc does not induce transcription in the presence of TFIIE and TFIIH, there is a pronounced increase in transcript formation in the absence of these two factors, especially for SNAPc-FL (Figure 1d, 1e). The effect observed in the presence of TFIIE and TFIIH could be the result of non-specific initiation (please see our response to reviewer #1). The effect of TFIIE alone increasing transcription of PIC has been well documented also in Pol II mRNA transcription on promoters with high meltability (Plaschka, Hantse et al 2016, Dienemann et al 2019). However, the lack of an additive effect of TFIIE and SNAPc together and the absence of TFIIE in the cryoEM structures, supports the non-cooperative role of TFIIE in Pol II snRNA transcription.

Final Decision Letter:

Dear Dr. Cramer,

We are now happy to accept your revised paper "Structural basis of SNAPc-dependent snRNA transcription initiation by RNA polymerase II" for publication as a Article in Nature Structural & Molecular Biology.

As soon as your article is published, you can generate your shareable link by entering the DOI of your article here: http://authors.springernature.com/share.

Corresponding authors will also receive an automated email with the shareable link

Note the policy of the journal on data deposition:

<http://www.nature.com/authors/policies/availability.html>.

Your paper will be published online soon after we receive proof corrections and will appear in print in the next available issue. You can find out your date of online publication by contacting the production team shortly after sending your proof corrections. Content is published online weekly on Mondays and Thursdays, and the embargo is set at 16:00 London time (GMT)/11:00 am US Eastern time (EST) on the day of publication. Now is the time to inform your Public Relations or Press Office about your paper, as they might be interested in promoting its publication. This will allow them time to prepare an accurate and satisfactory press release. Include your manuscript tracking number (NSMB-A46118A) and our journal name, which they will need when they contact our press office.

About one week before your paper is published online, we shall be distributing a press release to news organizations worldwide, which may very well include details of your work. We are happy for your institution or funding agency to prepare its own press release, but it must mention the embargo date and Nature Structural & Molecular Biology. If you or your Press Office have any enquiries in the meantime, please contact press@nature.com.

An online order form for reprints of your paper is available at https://www.nature.com/reprints/author-reprints.html. Please let your coauthors and your institutions' public affairs office know that they are also welcome to order reprints by this

method.

Please note that *Nature Structural & Molecular Biology* is a Transformative Journal (TJ). Authors may publish their research with us through the traditional subscription access route or make their paper immediately open access through payment of an article-processing charge (APC). Authors will not be required to make a final decision about access to their article until it has been accepted. [Find out more about Transformative Journals](https://www.springernature.com/gp/open-research/transformative-journals)

Sincerely,

Carolina Perdigoto, PhD
Chief Editor
Nature Structural & Molecular Biology
orcid.org/0000-0002-5783-7106
